# Brane detectors of a dynamical phase transition in a driven CFT

Suchetan Das[1][*], Bobby Ezhuthachan[2][†], Arnab Kundu[3,4][‡],
Somnath Porey[2][○], Baishali Roy[2][§] and Krishnendu Sengupta[5][¶]

**1** Department of Physics, Indian Institute of Technology Kanpur, Kanpur 208016, India.
**2** Ramakrishna Mission Vivekananda Educational and Research Institute,
Belur Math, Howrah-711202, West Bengal, India.
**3** Saha Institute of Nuclear Physics, 1/AF, Bidhannagar, Kolkata 700064, India.
**4** Homi Bhabha National Institute, Training School Complex,
Anushaktinagar, Mumbai 400094, India.
**5** School of Physical Sciences, Indian Association for the Cultivation of Science,
2A and 2B Raja S.C.Mullick Road, Jadavpur, Kolkata-700032, West Bengal, India.

[*] suchetan@iitk.ac.in , [†] bobby.phy@gm.rkmvu.ac.in , [‡] arnab.kundu@saha.ac.in ,
[○] somnathhimu00@gm.rkmvu.ac.in , [§] baishali.roy025@gm.rkmvu.ac.in , [¶] tpks@iacs.res.in

## Abstract

We show that a dynamical transition from a non-heating to a heating phase of a periodic $SL(2, \mathbb{R})$ driven two dimensional conformal field theory (CFT) with a large central charge is perceived as a first order transition by a bulk brane embedded in the dual AdS. We construct the dual bulk metric corresponding to a driven CFT for both the heating and the non-heating phases. These metrics are different $AdS_2$ slices of the pure $AdS_3$ metric. We embed a brane in the obtained dual AdS space and provide an explicit computation of its free energy both in the probe limit and for an end-of-world (EOW) brane taking into account its backreaction. Our analysis indicates a finite discontinuity in the first derivative of the brane free energy as one moves from the non-heating to the heating phase (by tuning the drive amplitude and/or frequency of the driven CFT) thus demonstrating the presence of the bulk first order transition. Interestingly, no such transition is perceived by the bulk in the absence of the brane. We also provide explicit computations of two-point, four-point out-of-time correlators (OTOC) using the bulk picture. Our analysis shows that the structure of these correlators in different phases match their counterparts computed in the driven CFT. We analyze the effect of multiple EOW branes in the bulk and discuss possible extensions of our work for richer geometries and branes.

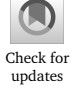

# 1 Introduction

Non-equilibrium dynamics of driven quantum matter has been extensively studied in the recent past [1–17]. Out of the several drive protocols that can be used to take a system out of equilibrium, periodic drives, whose stroboscopic dynamics can be described by Floquet Hamiltonians, have received the most attention [6, 9, 18–20]. The reason for this stems from several phenomena such as dynamical freezing [21–27], dynamical localization [28–33], topological transitions in driven systems [34–41], realization of time crystalline states [42–45], dynamical transitions [46–52], and tuning ergodicity properties of quantum systems [53, 54]; these phenomena have no analogue in either equilibrium or aperiodically driven non-equilibrium systems.

Several recent studies have focussed on the effect of both quench [55] and periodic drives [56–61] on conformal field theories. The studies involving periodic protocols usually consider a Hamiltonian which is expressed in terms of standard Virasoro generators $L_0$ and $L_{\pm 1}$ and is therefore valued in an $su(1,1)$-algebra. The periodic drive in such models leads to an evolution operator $U$ which is valued in SU(1,1). It is well-known that such a dynamics leads to two distinct phases separated by a dynamical transition. These are the heating (hyperbolic) and the non-heating (elliptic) phases; the Casimir of the $su(1,1)$ algebra has opposite signs in the two phases. The transition line where the Casimir vanishes is often referred to as the parabolic line. The presence of a periodic drive, characterized by a frequency $\omega_D = 2\pi/T$, where $T$ is the time period, allows one to access this dynamic transition by tuning the drive frequency. Equivalently, such a tuning is possible by changing the drive amplitude.

The AdS/CFT correspondence stipulates that corresponding to every such (1 + 1)D CFT, there exists a three-dimensional (3D) dual AdS bulk [62].[1] In fact, using this correspondence, one can have a definite procedure for extending the CFT Hamiltonian in the bulk [63]. It is then natural to ask what constitutes the bulk signature of the dynamic transition of a driven CFT. This is the central question which we aim to study in this work. More precisely, the main goal of our work is to geometrize the dynamical phase transition and to provide a precise and explicit 3D geometric and Holographic construction that captures this transition.

The main points of our work can be summarized as follows. First, we show that there are two key steps to construct the geometric description mentioned above. We take $CFT_2$ vacuum as the reference state which is dual to pure $AdS_3$. Since the vacuum does not evolve under $sl(2,R)$ valued Floquet-Hamiltonian, the bulk geometry remains pure AdS. However to geometrize the drive, the basic ingredient is the set of bulk generators corresponding to the Virasoro generators $L_0$ and $L_{\pm 1}$ of the boundary CFT [63]. Subsequently, one finds the curves in the bulk geometry which are generated by the bulk Hamiltonian corresponding to the the CFT Floquet-Hamiltonian. These curves inherit a natural induced metric on them, which are simply patches of the $AdS_3$-spacetime. These patches have a natural $AdS_2$ slicing, which differ for each phase. In particular, for the heating phase, we find a $AdS_2$ black hole slice. For the non-heating phase we find a global $AdS_2$ slicing and the phase boundary corresponds to a Poincare $AdS_2$ slicing. While these patches are highly suggestive, the heating and the non-heating phases cannot be distinguished by the corresponding free energy, which is given by the Euclidean on-shell action for pure 3D gravity in $AdS_3$. At this point, it is worth emphasizing that the boundary CFT Hamiltonian is $sl(2,R)$-valued and therefore there are no large gauge transformations in the bulk. Thus, the heating and the non-heating phases are identical in terms of their Euclidean on-shell actions.

Second, we find that the crucial ingredient in distinguishing between these CFT phases is a brane degree of freedom. These branes are co-dimension one hypersurfaces in the bulk geometry. In this work, we have considered both probe branes as well as end-of-world (EOW) branes. These are respectively probes and fully back-reacting objects in the $AdS_3$ geometry. Given a particular patch (corresponding to the heating, the non-heating or the phase boundary), these branes can distinguish between the on-shell Euclidean action of the (gravity + brane)-system. The central result of this work is that the corresponding free energy displays a first order phase transition of the combined system, which is a close cousin of the Hawking-Page transition [64]. We explicitly demonstrate that this first order transition also occurs due to the change in sign of the Casimir of the boundary CFT, thereby establishing a direct link between it and the dynamical transition of the driven CFT mentioned above.

Third, we generalize this construction and introduce more than one EOW-branes. As an example, we have considered two EOW-branes, which results in a rich structure associated to the phase transition. The basic qualitative features of the phase transition remain the same. We note that insertion of the EOW-branes correspond to inserting conformal boundaries to the boundary CFT. Correspondingly, the CFT is defined on a strip with an infinite family of boundary conditions. These boundary conditions are labelled by the respective brane tensions valued in the range of [0, 1]. Especially, this family of boundary conditions naturally allow for exciting a boundary-condition-changing operator in the CFT, whenever the boundary conditions are non-identical at the end points.

Finally, we also study the signature of different phases in unequal time two point and higher point correlators under the $sl(2,R)$ drive [61,65] from the bulk gravity picture without inserting any brane. In this context, we describe how the different $AdS_2$ slicing corresponding

---

[1]Note that the CFT dual of pure 3D gravity is a debated issue. A precise duality generally contains various fluxes in a 10D bulk geometry. However, we will ignore this issue for now, as we will be focussing on a rather generic point which is expected to remain qualitatively true.

to the two phases and the phase boundary are crucial in determining different temporal behavior of 2-point and 4-point functions from the Holographic description, which can be matched with results available from large $c$ CFT computations. In particular, we match the two point functions in each phase with a direct boundary computation of the two point function. We then set-up the OTOC computation in the bulk. This involves, the by now well-known method of, computing a two point function of an operator in a shock wave geometry created by the other operator [66, 67]. The AdS$_2$ black hole slicing for the heating phase is crucial to obtain the exponential temporal growth of the OTOC,[2] that we had obtained in our previous work [65]. We show that the Lyapunov exponent matches exactly with the boundary computation. Moreover we also show how the other AdS$_2$ slicing corresponding to the non heating phase and the phase boundary results in an oscillatory and power law temporal growth of the OTOCs in these two phases.

The plan for the rest of the paper is as follows. In Sec. 2, we construct the bulk metric for the different phases of the driven CFTs. This is followed by Sec. 3, where we discuss embedding both probe and EOW branes. Next, in Sec. 4, we compute two-point correlation functions and four-point OTOC from the bulk in the large $c$ limit and compare these results with the corresponding counterparts obtained from the driven CFT at the boundary. Finally, we discuss our main results and the possibilities of their further extension and conclude in Sec. 5.

## 2 Bulk metrics in different phases subjected to an $SL(2, \mathbb{R})$ drive

In this section, we construct the bulk metrics for the various phases of the $SL(2, \mathbb{R})$ driven CFT.

**General strategy:** To begin with let us consider a generic state $|\psi\rangle$ that is evolved in stroboscopic time $n$ under some 2D boundary periodically driven Hamiltonian $H$. We want to understand the three dimensional holographic realization of the state as well as it's evolution i.e.

$$|\psi(n)\rangle = U(nT, 0)|\psi\rangle = e^{-iH_F nT/\hbar}|\psi\rangle, \tag{1}$$

where $U(nT, 0)$ is the evolution operator and $H_F$ is the Floquet Hamiltonian. Here $n$ is a positive integer, $T = 2\pi/\omega_D$ is the drive period, and $\omega_D$ is the drive frequency. The complete holographic picture could be obtained by a two step process:

- First we find the geometric dual of a one parameter class of states of the above form, with $nT$ replaced by $s$. At this step, $s$ (or rather $s/T$) should be interpreted as a real parameter of the geometric description. The geometric dual can then be found by solving the Einstein equation with source given by the expectation value of boundary stress tensor in certain choice of coordinate system.

- The next step is to rewrite the new metric in a parametrization where $s$ itself becomes the time in the metric. This means going to the frame of the co-moving observer along the curve generated by the Floquet Hamiltonian itself. The well-known example is the Rindler wedge which is obtained by changing the flat spacetime coordinates to new coordinates generated by the trajectory of an accelerated observer. Once we are able to get the effective boundary metric by parametrizing the boundary curve, we need to lift that into the bulk [68–71]. One straightforward yet harder way to get the final bulk metric is to solve the same Einstein equation as in the first step with boundary metric as

---

[2]We want to emphasize again that the dual 3d AdS metric has no horizon and the blackhole resides in it's AdS$_2$ slice. To the best of our knowledge, OTOCs in these kinds of geometries have not been studied earlier.

the boundary condition. However, our task is simpler: We can directly solve for the bulk curves generated by the bulk representation of the boundary Floquet Hamiltonian. The curves will be parametrized by $s$ and other intrinsic co-ordinates, in terms of which one rewrites the above metric.

The simplest example of the above set up is when we take $H = H_{\text{CFT}} = L_0 + \bar{L}_0$ and the state is the vacuum $|0\rangle$. The state does not evolve in (the stroboscopic) time as $e^{i(L_0 + \bar{L}_0)n}|0\rangle = |0\rangle$. In the Euclidean boundary, this corresponds to radial quantization which can be visualized by conformally mapping the plane into a cylinder, where $n$ coincides with the time direction in the cylinder. On the bulk, this corresponds to global AdS$_3$ with $n$ naturally enlarged to $s$ which acts as the global time. If we choose $H$ to be other linear combinations of conformal generators $L_p, \bar{L}_p$, it corresponds to a different quantization [72], [73] in the CFT. The corresponding bulk metric will be obtained by mapping it from the AdS$_3$ coordinates under large diffeomorphism (generated by boundary $L_p$'s) in a specific gauge [74] and then solving for the bulk curve. Let us now discuss this explicitly when $p = \{0, \pm 1\}$.

**Bulk metric under an $SL(2, \mathbb{R})$ drive:**    To compute the bulk metric in an $SL(2, \mathbb{R})$ driven CFT, we extend the boundary Hamiltonian into the bulk by replacing the global Virasoro generators by it's AdS$_3$ representation [63]:

$$L_{b,0} = -\frac{1}{2}z\partial_z - \zeta\partial_\zeta, \quad L_{b,1} = \frac{1}{2}z\zeta\partial_z + \zeta^2\partial_\zeta - z^2\partial_{\bar{\zeta}}, \quad L_{b,-1} = \partial_\zeta. \tag{2}$$

Here, $\zeta = x - i\tau$ and $\bar{\zeta} = x + i\tau$ are the boundary coordinates and $z$ is along the bulk direction. We will work in two step discrete drive protocol [56–58] governed by the Hamiltonian $H_\phi = L_0 - \frac{1}{2}\tanh(2\phi)(L_1 + L_{-1}) +$ anti chiral part. For time period $\tau_0$ the system is evolved by $H_0 = H_{\phi=0}$, and for time period $\tau_1$ it is evolved by $H_1 = H_{\phi\neq0}$ and then we repeat it periodically for $n$ number of drives. The stroboscopic time parameter $n$ plays the role of time in our setting. Here we still start with the vacuum $|0\rangle$. Since again $H_\phi$ is constructed out of $SL(2, \mathbb{R})$ generators, the vacuum remains unchanged. This dictates that the bulk remains pure AdS$_3$. However to construct the bulk tangent curve along the direction of drive $n$, it might be difficult to track down the time dependent set up at each period of time. For computational purposes, it would be useful to find the Floquet Hamiltonian which controls the evolution of the driven system after an integer number of drive periods. We can write an effective Hamiltonian with the following form

$$H_{\text{eff}} = \alpha\left(L_0 + \bar{L}_0\right) + \beta\left(L_1 + \bar{L}_1\right) + \gamma\left(L_{-1} + \bar{L}_{-1}\right). \tag{3}$$

We give the relevant details of $\alpha, \beta, \gamma$ in the appendix A. The corresponding AdS$_3$ representation of the Hamiltonian is given by,

$$H_b = \alpha\left(L_{b,0} + \bar{L}_{b,0}\right) + \beta\left(L_{b,1} + \bar{L}_{b,1}\right) + \gamma\left(L_{b,-1} + \bar{L}_{b,-1}\right). \tag{4}$$

This class of Hamiltonians are valued in $su(1,1)$-algebra which generates a time-evolution valued in the $SU(1,1)$ group. Given $\{\alpha, \beta, \gamma\}$, the Casimir of the algebra is given by $(\alpha^2 - 4\beta\gamma)$ and we define:

$$d = \frac{\alpha^2 - 4\beta\gamma}{4\beta^2}, \tag{5}$$

which keeps track of the sign of the Casimir. It is now well-known that the system can be tuned to any of the three distinct phases, depending on the sign of $d$, see *e.g.* [56–61] for several related works exploring these phases.

Specifically, we can distinguish the three phases of the system:

$$d < 0: \quad \text{Heating Phase},$$
$$d > 0: \quad \text{Non-heating Phase},$$
$$d = 0: \quad \text{Phase transition}.$$

Substituting (2) and corresponding complex conjugates in (4) we can write:

$$H_b = (-\alpha z + 2\beta z x)\partial_z + (-\alpha x - \beta z^2 + \beta(x^2 - \tau^2) + \gamma)\partial_x + (-\alpha\tau + 2\beta x\tau)\partial_\tau. \quad (6)$$

As mentioned before, we consider an intrinsic coordinate, denoted by $s \in \mathbb{R}$, to parameterize the curves generated by the bulk Hamiltonian and solve the following tangent equations [75]:

$$\frac{dz(s)}{ds} = -\alpha z + 2\beta z x, \quad (7)$$

$$\frac{d\tau(s)}{ds} = -\alpha\tau + 2\beta x\tau, \quad (8)$$

$$\frac{dx(s)}{ds} = -\alpha x - \beta z^2 + \beta(x^2 - \tau^2) + \gamma, \quad (9)$$

to find the relation between the embedding coordinates $\{\tau, x, z\}$ and the patch solved by the equations in (7-9). Note that, the bulk coordinate $s$ is continuous while the stroboscopic time is discrete. The identification of the stroboscopic time with this continuous bulk coordinate is made only at discrete points. Said another way, different values of the stroboscopic time correspond to different points on the curve whose coordinate is $s$. Note also, that a solution to the above equations will allow an arbitrary constant shift in $s$ and therefore, effectively we can set the range: $s \in [-\infty, \infty]$. The solution space can be divided into three categories, depending on the sign of $d$. Below, we discuss these in detail.

**For non-heating phase($d > 0$):**

When $d > 0$, the set of equations in (7)-(9) is solved by

$$x = -\frac{\sqrt{d}}{2}\Big(\coth[\mu(s + i\theta)] + \coth[\mu(s - i\theta)]\Big),$$

$$\tau = -\frac{\sqrt{d}}{2i\sqrt{1 + c_1^2}}\Big(\coth[\mu(s + i\theta)] - \coth[\mu(s - i\theta)]\Big), \quad (10)$$

$$z = -\frac{\sqrt{d}\,c_1}{2i\sqrt{1 + c_1^2}}\Big(\coth[\mu(s + i\theta)] - \coth[\mu(s - i\theta)]\Big), \quad \mu = \beta\sqrt{d}. \quad (11)$$

Here $c_1$ and $\theta$ characterize the parametric solutions. After rewriting $c_1 = \tan\phi_1$ and substituting (10) in AdS$_3$-Poincaré metric, $ds^2 = \frac{dx^2 + d\tau^2 + dz^2}{z^2}$ we get[3]

$$ds^2 = \frac{d\phi^2}{\sin^2[\phi]} + \frac{4\beta^2 d\,(ds^2 + d\theta^2)}{\sin^2[\phi]\sin^2[2\sqrt{d}\beta\theta]}. \quad (12)$$

Finally, analytically continuing $s \to is$ we obtain:

$$ds^2 = \frac{d\phi^2}{\sin^2[\phi]} + \frac{4\beta^2 d\,(-ds^2 + d\theta^2)}{\sin^2[\phi]\sin^2[2\sqrt{d}\beta\theta]}. \quad (13)$$

---

[3]When we are substituting in the metric, we treat $\theta$ and $\phi$ to be the normal coordinates to the curve.

The ranges of coordinates $s, \theta, \phi$ are respectively given by $[-\infty, +\infty], [0, \frac{\pi}{\mu}], [0, +\pi]$. It is straightforward to check that these ranges cover the full Poincaré patch of AdS$_3$, i.e. $\tau \in [-\infty, \infty]$, $x \in [-\infty, \infty]$ and $z \in [0, \infty]$. The metric in (13) describes an AdS$_3$ foliated by AdS$_2$ geometries at each $\phi = $ const. It is instructive to note that, by comparing (13) with eqn (3.1) in [76], the constant $\phi$ slices correspond to global-AdS$_2$ geometry. We will revisit this in detail later.

**For heating phase (d< 0):**

When $d < 0$, the set of equations in (7)-(9) is solved by

$$
\begin{aligned}
x &= \frac{\sqrt{d}}{2}\Big(\tan[\mu(s+i\theta)] + \tan[\mu(s-i\theta)]\Big), \\
\tau &= \frac{\sqrt{d}}{2i\sqrt{1+c_1^2}}\Big(\tan[\mu(s+i\theta)] - \tan[\mu(s-i\theta)]\Big), \\
z &= \frac{\sqrt{d}\,c_1}{2i\sqrt{1+c_1^2}}\Big(\tan[\mu(s+i\theta)] - \tan[\mu(s-i\theta)]\Big).
\end{aligned}
\tag{14}
$$

As before, substituting (14) in AdS$_3$-Poincaré metric, $ds^2 = \frac{dx^2 + d\tau^2 + dz^2}{z^2}$ we get:

$$
ds^2 = \frac{d\phi^2}{\sin^2[\phi]} + \frac{4\beta^2 d\,(ds^2 + d\theta^2)}{\sin^2[\phi]\sinh^2[2\sqrt{d}\beta\theta]}.
\tag{15}
$$

Again, the analytic continuation: $s \to is$ gives

$$
ds^2 = \frac{d\phi^2}{\sin^2[\phi]} + \frac{4\beta^2 d\,(-ds^2 + d\theta^2)}{\sin^2[\phi]\sinh^2[2\sqrt{d}\beta\theta]}.
\tag{16}
$$

The ranges of variables $s, \theta, \phi$, for the metric (16) in heating phase are given by $(-\infty, +\infty), (0, \infty), (0, \pi)$ respectively. The $\phi = $ const slices of (16) are now AdS$_2$ black holes which is explicitly visible by comparing (16) with equation (3.3) in [76].

**On the transition line (d= 0):**

By using exact similar analysis for $d = 0$, the coordinates and corresponding analytically continued metric can be written down as follows:

$$
x = -\frac{1}{2\beta}\Big(\frac{1}{s+i\theta} + \frac{1}{s-i\theta}\Big), \quad \tau = -\frac{1}{2i\beta\sqrt{1+c_1^2}}\Big(\frac{1}{s+i\theta} - \frac{1}{s-i\theta}\Big),
$$
$$
z = -\frac{c_1}{2i\beta\sqrt{1+c_1^2}}\Big(\frac{1}{s+i\theta} - \frac{1}{s-i\theta}\Big).
\tag{17}
$$

This yields:

$$
ds^2 = \frac{d\phi^2}{\sin^2[\phi]} + \frac{-ds^2 + d\theta^2}{\sin^2[\phi]\,\theta^2}.
\tag{18}
$$

In this case, the $\phi = $ const slices corresponds to the AdS$_2$-Poincaré patch [76]. We will now discuss how these patches determine the physics of the transition, especially by inserting explicit brane degrees of freedom inside the bulk geometry.

# 3 Brane embeddings in AdS$_3$

In this section we will demonstrate how a non-trivial conformal boundary can detect the phase transition. Our explicit calculations will be carried out in the Holographic description, since it provides us with a natural and simple way to characterize various boundary conditions on the conformal boundaries of the CFT. In the Holographic dual, such boundaries correspond to defect branes which are described by hypersurfaces in the geometry. We will show below that these branes can detect the heating to non-heating phase transition in both a probe limit as well as away from the probe limit. Before proceeding further, let us recall that the relevant metric data, in the Euclidean description, are given by

$$ds^2 = \frac{d\phi^2}{\sin^2[\phi]} + \frac{4\mu^2 \, (ds^2 + d\theta^2)}{\sin^2[\phi] \sin^2[2\mu\theta]}, \quad \mu = \beta \sqrt{|d|}, \quad d > 0, \tag{19}$$

$$s \in [-\infty, \infty], \quad \theta \in \left[0, \frac{\pi}{\mu}\right], \quad \phi \in [0, \pi], \tag{20}$$

for the non-heating phase. Similarly, for the heating phase, we obtain:

$$ds^2 = \frac{d\phi^2}{\sin^2[\phi]} + \frac{4\mu^2 \, (ds^2 + d\theta^2)}{\sin^2[\phi] \sinh^2[2\mu\theta]}, \quad \mu = \beta \sqrt{|d|}, \quad d < 0, \tag{21}$$

$$s \in [-\infty, \infty], \quad \theta \in [0, \infty], \quad \phi \in [0, \pi]. \tag{22}$$

Note that, in both (19) and (21), we can absorb the factor of $\mu$ by redefining $s \to 2\mu s$ and $\theta \to 2\mu\theta$ and the resulting metric becomes independent of $\mu$. However, the geometries retain the memory of sgn($d$) since (21) is obtained by sending $\mu \to -i\mu$ (equivalent to sending $d \to -d$) in (19).[4] In the subsequent discussions, we will keep the factor of $\mu$ explicit.

## 3.1 A Lorentzian discussion

It is evident from (19) and (21) that the $\phi = $ const slices are special. This will prove crucial in the subsequent discussions and here we will discuss the Lorentzian picture in some detail, which will form the basic intuition in all subsequent observations. The Lorentzian patches are obtained by sending $s \to is$ on the $\phi = \phi_0$ slices. The induced metric on the various phases are:

$$ds^2 = \frac{4\mu^2}{\sin^2 \phi_0} \left( \frac{-ds^2 + d\theta^2}{\sin^2(2\mu\theta)} \right), \quad s \in [-\infty, \infty], \quad 2\mu\theta \in [0, \pi], \tag{23}$$

$$ds^2 = \frac{4\mu^2}{\sin^2 \phi_0} \left( \frac{-dt^2 + d\xi^2}{\sinh^2(2\mu\xi)} \right), \quad t \in [-\infty, \infty], \quad \xi \in [0, \infty], \tag{24}$$

$$ds^2 = \frac{4\mu^2}{\sin^2 \phi_0} \left( \frac{-dT^2 + dX^2}{X^2} \right), \quad T \in [-\infty, \infty], \quad X \in [0, \infty]. \tag{25}$$

Here (23), (24) and (25) correspond to non-heating, heating phases and at the transition point. These patches describe various parts of an AdS$_2$ geometry. Making explicit use of these metrics in [76], they are also related to each other by simple co-ordinate transformations. Explicitly,

$$T + X = \tan\left( \frac{s + 2\mu\theta}{2} \right), \quad T - X = \tan\left( \frac{s - 2\mu\theta}{2} \right), \tag{26}$$

$$\tan\left( \frac{s - \frac{\pi}{2} + 2\mu\theta}{2} \right) = -e^{-2\mu(t+\xi)}, \quad \tan\left( \frac{s + \frac{\pi}{2} - 2\mu\theta}{2} \right) = e^{2\mu(t-\xi)}, \tag{27}$$

---

[4]Recall that the phase transition takes place as a function of sgn($d$), which is the sign of the Casimir of the $SU(1,1)$ evolution in the driven CFT.

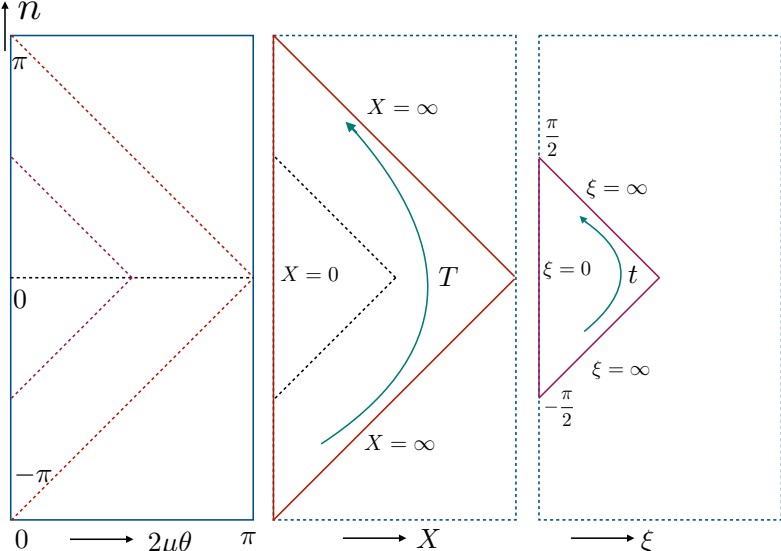

Figure 1: A pictorial representation of various patches of $AdS_2$ covered by various phases. The left-most is the non-heating phase that covers the global patch of $AdS_2$, the middle one covers the Poincaré patch of $AdS_2$ and the right most covers a Schwarzschild-like patch in $AdS_2$. These patches are explicitly related by the coordinate transformations in (26) and (27).

which relate the non-heating patch to the transition patch in (26) and the non-heating patch to the heating patch in (27). Note from (26) that the line $s = 2\mu\theta - \pi$ and $s = \pi - 2\mu\theta$ both map to $X = \infty$. On the other hand, $\theta = 0$ is mapped to $X = 0$. Similarly, it is straightforward to check that $s = 2\mu\theta - \pi/2$ and $s = \pi/2 - 2\mu\theta$ map to $\xi = \infty$ line, while $\theta = 0$ maps to $\xi = 0$ line. These patches are pictorially represented in Fig. 1. These $\phi = $ const $AdS_2$ patches will be crucial in the subsequent sections.

## 3.2 Probe branes

Let us first consider probing the geometries in (19) and (21) with a brane.[5] Consider a two-dimensional hypersurface with the following action:

$$S_{\text{brane}} = T \int d^2\sigma \sqrt{\gamma} = T \int d^2\sigma \mathcal{L}, \tag{28}$$

where, $\gamma_{ab} = g_{\mu\nu}\partial_a X^\mu \partial_b X^\nu$, is the induced metric and, $g_{\mu\nu}$ is the background metric and $T$ is the tension in the brane. Let us choose the world-volume coordinates to be: $\sigma^0 = s$, $\sigma^1 = \theta$, and let $\phi(\theta)$ denote the corresponding embedding function. The corresponding induced metrics are:

$$ds^2 = \frac{1}{\sin^2\phi(\theta)}\left(\frac{4\mu^2 ds^2}{\sinh^2(2\mu\theta)} + \left(\phi'^2 + \frac{4\mu^2}{\sinh^2(2\mu\theta)}\right)d\theta^2\right), \quad \text{heating}, \tag{29}$$

$$ds^2 = \frac{1}{\sin^2\phi(\theta)}\left(\frac{4\mu^2 ds^2}{\sin^2(2\mu\theta)} + \left(\phi'^2 + \frac{4\mu^2}{\sin^2(2\mu\theta)}\right)d\theta^2\right), \quad \text{non-heating}. \tag{30}$$

---

[5]Note that probe branes can be used to detect phase transitions across a wide range of systems, *e.g.* [77], [78], [79].

With the above ansatz, the Lagrangian becomes a functional of the embedding function $\mathcal{L} = \mathcal{L}[\theta, \phi, \phi']$ and the brane profile can be obtained by solving the Euler-Lagrange equation:

$$\frac{d}{d\theta}\left(\frac{\partial \mathcal{L}}{\partial \phi'}\right) - \frac{\partial \mathcal{L}}{\partial \phi} = 0. \tag{31}$$

It is straightforward to observe that in both phases, the Euler-Lagrange equation admits a simple, analytical solution $\phi(\theta) = \pi/2$.[6] For analytical control on the calculations, we will discuss only this solution. The corresponding on-shell actions of the probe branes, in both phases, can be computed by substituting this solution into the action. This yields:

$$S_{\text{brane}}^{\text{heating}} = T \int ds\, d\theta\, \frac{4\mu^2}{\sinh^2(2\mu\theta)} = -2\mu T \coth(2\mu\theta)\Big|_{\theta_{\min}}^{\theta_{\max}} \int ds$$
$$= \int ds\left(-2\mu T + \frac{T}{\epsilon_{\text{h}}}\right), \tag{32}$$

$$S_{\text{brane}}^{\text{non-heating}} = T \int ds\, d\theta\, \frac{4\mu^2}{\sin^2(2\mu\theta)} = -2\mu T \cot(2\mu\theta)\Big|_{\theta_{\min}}^{\theta_{\max}} \int ds$$
$$= \int ds\left(\frac{2T}{\epsilon_{\text{nh}}}\right). \tag{33}$$

It is straightforward to observe that by choosing $\epsilon_{\text{h}} = \epsilon_{\text{nh}}/2$ the divergent pieces in the heating and the non-heating phases become equal. Here, to regulate the divergences, we have introduced two cut-offs $\epsilon_{\text{h}} = \theta_{\min}$ in the heating phase, and $\epsilon_{\text{nh}} = \theta_{\min} = \pi/(2\mu) - \theta_{\max}$ in the non-heating phase.

The phase transition can be detected by considering the difference in their respective on-shell actions: $\Delta S = S_{\text{brane}}^{\text{heating}} - S_{\text{brane}}^{\text{non-heating}}$. This is formally divergent, unless we choose $\epsilon_{\text{h}} = \epsilon_{\text{nh}}/2$. This is certainly an allowed choice and it yields: $\Delta S \sim -2\mu T < 0, \forall\, T > 0$. Alternatively, we can renormalize the corresponding on-shell actions by adding appropriate counter-terms to the respective branes. In the non-heating phase, there are two boundaries: $\theta \to 0$ and $\pi/(2\mu) - \theta \to 0$, while the heating phase has only one boundary limit $\theta \to 0$. The corresponding on-shell action can be renormalized by introducing the following boundary terms:

$$S_{\text{ct}}^{\text{non-heating}} = \int ds\, \sqrt{h}\,\Big|_{\theta = \frac{\pi}{\mu} - \epsilon_{\text{nh}}} - \int ds\, \sqrt{h}\,\Big|_{\theta = \epsilon_{\text{nh}}}, \tag{34}$$

$$S_{\text{ct}}^{\text{heating}} = -\int ds\, \sqrt{h}\,\Big|_{\theta = \epsilon_{\text{h}}}, \tag{35}$$

such that $S_{\text{brane}}^{\text{heating}} + S_{\text{ct}}^{\text{heating}}$ and $S_{\text{brane}}^{\text{non-heating}} + S_{\text{ct}}^{\text{non-heating}}$ are both finite. Here $h$ denote the induced metric on the boundary (i.e. $\theta = \text{const}$ slice) of the brane.

Several comments are in order. First, it is clear that for a fixed tension brane, the free energy is lowered as $\text{sgn}(d)$ crosses zero from the positive side. We emphasize again that even though the factor of $(\mu T)$ can be absorbed in redefining $s$, the memory of $\text{sgn}(d)$ remains in the final answer. Here $\text{sgn}(d)$ corresponds to the sign of the Casimir that distinguishes between the non-heating and the heating phases. This phase transition is a first order one, since it is straightforward to observe that e.g. $(\partial S/\partial \mu)$ have a discontinuous jump at the transition.[7]

---

[6]There is a family of solutions to the Euler-Lagrange equation, subject to appropriate boundary conditions, which can be obtained numerically.

[7]Also note that, the free energy does not have the detailed swallow-tail structure associated with a typical first order phase transition. This is perhaps due to the simplicity of the system.

Intuitively, one would prefer the positive tension branch, since it corresponds to a positive kinetic energy for the brane and satisfy standard positive energy conditions on the brane. The negative tension, on the other hand, corresponds to a negative kinetic energy and can lead to instabilities. Nonetheless, such objects appear naturally within the context of string theory as *e.g.* orientifold planes (see *e.g.* [80]) and can play pivotal role in realizing interesting cosmological scenario.

## 3.3 End-of-world (EOW) branes

We will now consider introducing fully back-reacting and dynamical End-of-World (EOW) branes in the corresponding heating and non-heating patches of AdS$_3$ geometry.[8] Since the two patches are related by local co-ordinate transformations and not by large gauge transformations, the on-shell action of the three-dimensional Einstein-Hilbert term along with the Gibbons-Hawking boundary term cannot distinguish between the two phases. An EOW brane introduces a hypersurface dynamics which is determined by the extrinsic curvature and is therefore not a topological quantity in two-dimensions. Thus, it is expected that the phase transition will be explicitly visible once EOW-branes are inserted into the geometry. We will first consider a single EOW-brane and subsequently discuss two EOW-branes.

### 3.3.1 Single EOW brane

The full bulk action now comprises of several pieces: The gravity part, the brane part and the intersection boundary part between the bulk geometry and the brane:

$$S_{\text{full}} = S_{\text{gravity}} + S_{\text{brane}} + S_{\text{corner}}, \tag{36}$$

$$S_{\text{gravity}} = -\frac{1}{2\kappa^2} \int_{\mathcal{M}} d^3x \sqrt{g} \, (R - 2\Lambda) - \frac{1}{\kappa^2} \int_{\partial\mathcal{M}} d^2y \sqrt{h} K, \tag{37}$$

$$S_{\text{brane}} = -\frac{1}{\kappa^2} \int_{\Sigma} d^2\sigma \sqrt{\gamma} \, (K - T), \tag{38}$$

$$S_{\text{corner}} = -\frac{1}{\kappa^2} \int_{\mathcal{C}} d\xi \sqrt{h_{\mathcal{C}}} \left( \pi - \Theta_{\Sigma, \partial\mathcal{M}} \right), \quad \mathcal{C} = \Sigma \cap \partial\mathcal{M}. \tag{39}$$

Here $d^3x$, $d^2y$, $d^2\sigma$ and $d\xi$ denote the volume element of the full bulk geometry, the conformal boundary, the brane and the corner. Correspondingly, $g$, $h$, $\gamma$ and $h_{\mathcal{C}}$ denote the metrics on them, $K$ denotes the corresponding extrinsic curvatures and $T$ is the brane tension. The angle $\Theta_{\Sigma, \partial\mathcal{M}}$ denote the angle at which the brane intersects the conformal boundary. The variational problem on (36) is defined by varying the inverse metric within the region bounded by the branes and the boundary, keeping the branes and the corners fixed. This yields the following equations:

$$R_{\mu\nu} - \frac{1}{2} (R - 2\Lambda) g_{\mu\nu} = 0, \tag{40}$$

$$K_{ab} - (K - T) \gamma_{ab} = 0. \tag{41}$$

The first equation (*i.e.* Einstein equations) above determines the three-dimensional bulk geometry and the second equation determines the profile of the brane. For us, the Einstein equations are satisfied simply because we consider an AdS$_3$ geometry.

---

[8]Note that our framework will be very similar to *e.g.* [81–83], where unrelated physics questions have been addressed.

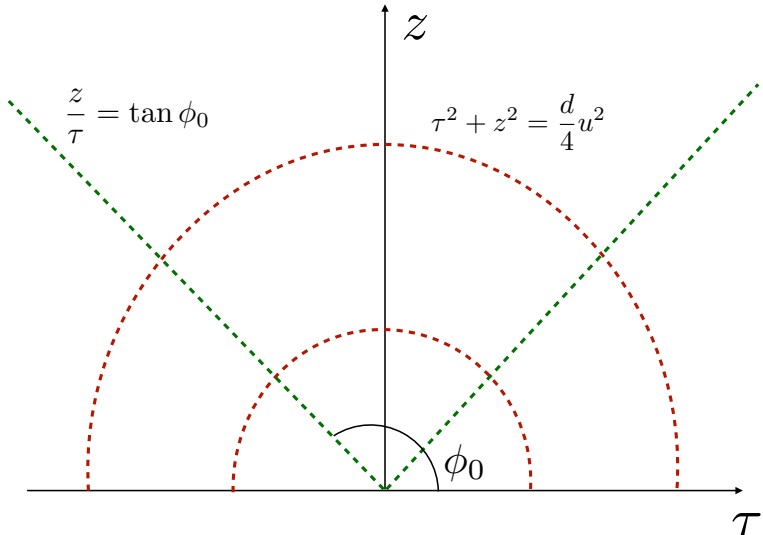

Figure 2: A pictorial representation of the two types of foliations described by equations (46) and (47). For the linear leaves, positive tension Karch-Randall branes are located at constant angle $\phi_0 \in [\frac{\pi}{2}, \pi]$ and for negative tension, they correspond to $\phi_0 \in [0, \frac{\pi}{2}]$. Here, the EOW-brane intersects the conformal boundary at $\tau = 0$.

Before proceeding further, let us note that the non-heating patch is described by

$$x = -\frac{\sqrt{|d|}}{2} v, \qquad \tau = -\frac{\sqrt{|d|}}{2} u \cos\phi, \qquad z = -\frac{\sqrt{|d|}}{2} u \sin\phi, \tag{42}$$

$$u = \frac{1}{i}\left[\coth\big(\mu(s+i\theta)\big) - \coth\big(\mu(s-i\theta)\big)\right], \quad v = \left[\coth\big(\mu(s+i\theta)\big) + \coth\big(\mu(s-i\theta)\big)\right]. \tag{43}$$

Similarly, the heating phase is described by the following patch:

$$x = -\frac{\sqrt{|d|}}{2} v, \quad \tau = \frac{\sqrt{|d|}}{2} u \cos\phi, \quad z = \frac{\sqrt{|d|}}{2} u \sin\phi, \tag{44}$$

$$u = \frac{1}{i}\left[\tan\big(\mu(s+i\theta)\big) - \tan\big(\mu(s-i\theta)\big)\right], \quad v = \left[\tan\big(\mu(s+i\theta)\big) + \tan\big(\mu(s-i\theta)\big)\right]. \tag{45}$$

Both the patches are essentially described by the following equations:

$$\tau^2 + z^2 = \frac{d}{4} u^2, \tag{46}$$

$$\frac{z}{\tau} = \tan\phi. \tag{47}$$

From (46), we observe that each $u = $ const describes a leaf of a circular foliation of the Poincaré patch and (47) implies that each $\phi = $ const describes a leaf of a planar foliation of the same. It is expected that each leaf of both the planar and the circular foliation is described by a Karch-Randall brane of a given tension. We will explicitly show this and for simplicity, we will focus on the planar foliations.

Let us choose $\sigma^0 = s$ and $\sigma^1 = \theta$ as the worldvolume coordinates, and $\phi = \phi(\theta)$ as the embedding function. This implies: $d\phi - \phi' d\theta = 0$ and therefore the unit outward normal to

the brane is given by

$$n_\alpha^{\text{nh}} = \frac{2\mu}{\sin\phi} \frac{1}{\sqrt{4\mu^2 + \phi'^2 \sin^2(2\mu\theta)}} (0, -\phi', 1),$$ (48)

$$n_\alpha^{\text{h}} = \frac{2\mu}{\sin\phi} \frac{1}{\sqrt{4\mu^2 + \phi'^2 \sinh^2(2\mu\theta)}} (0, -\phi', 1),$$ (49)

where $n_\alpha^{\text{nh}}$ and $n_\alpha^{\text{h}}$ correspond to non-heating and heating phases, respectively. The extrinsic curvature is calculated by using $K_{ab} = \nabla_\alpha n_\nu e_a^\alpha e_b^\nu$, where $e_a^\alpha = (\partial x^\alpha / \partial \sigma^a)$. The simplest component of the brane equation in (41) is given by the $nn$-component which can be readily solved to obtain the profile: $\phi(\theta) = \phi_0$, where $T = -\cos\phi_0$, in both phases. It is now straightforward to check that this solves the full equations in (41).

Let us now evaluate the on-shell actions in the corresponding phases. First, in the non-heating phase, we obtain:

$$S_{\text{gravity}} = -\frac{1}{\kappa^2} \frac{1}{\epsilon^2} \int ds \int_0^{\pi/2\mu} \frac{d\theta}{\sin^2(2\mu\theta)}, \quad S_{\text{brane}} = -\frac{1}{\kappa^2} \frac{2}{\epsilon} \frac{T}{1-T^2} \int ds,$$ (50)

$$S_{\text{corner}} = -\frac{1}{\kappa^2} \phi_0 \frac{1}{\epsilon} \int ds.$$ (51)

We add the following counter-term:

$$S_{\text{ct}} = \frac{1}{\kappa^2} \left( \frac{2T}{1-T^2} + (\pi - \phi_0) \right) \int d\xi \sqrt{h} \Bigg|_{\theta=\epsilon} + A_{\text{nh}} \frac{1}{\kappa^2} \int \sqrt{h} \Bigg|_{\phi=\pi-\epsilon},$$ (52)

$$A_{\text{nh}} = \int dn \int_0^{\pi/2\mu} \frac{d\theta}{\sin^2(2\mu\theta)}.$$ (53)

Here $h$ denotes the induced metrics at the corresponding hyper-surfaces. Note that the $z = \epsilon$ hypersurface corresponds to $\phi = \pi - \epsilon$ hypersurface. Also note that, the coefficient $A_{\text{nh}}$ is formally a divergent quantity, which itself needs a regularization. Nonetheless, the upshot is that there is no finite contribution from the counter-terms and therefore the sum of $S_{\text{full}} + S_{\text{ct}} = 0$, in the non-heating phase.

A similar computation in the heating phase require identical counter-terms as above, except $A_{\text{nh}} \to A_{\text{h}}$, where

$$A_{\text{h}} = \int ds \int_0^\infty \frac{d\theta}{\sinh^2(2\mu\theta)}.$$ (54)

Note that, while $A_{\text{h}}$ is still formally a divergent quantity and needs regularization, the $\theta$-integral produces a finite contribution as $\theta \to \infty$. This very feature becomes crucial on the brane. In the non-heating phase, the brane on-shell action consists only of divergent contributions while in the heating phase the $\theta$-integral contains a finite piece, as we just noticed. Upon introducing the counter-terms this finite contribution survives and we obtain:

$$S_{\text{full}} + S_{\text{ct}} = \frac{1}{\kappa^2} \frac{2\mu T}{1-T^2} \int ds.$$ (55)

A few comments are in order. Note that, in the tension-less limit $T \to 0$ and therefore $\phi_0 \to \pi/2$, which recovers the probe limit answer of equation (31). In the strict $T = 0$ limit, (55) vanishes, which is also consistent with the probe limit calculation. The free energy

in the small tension limit, however, does not reduce to the probe limit answer since the extrinsic curvature still contributes to the full action. This is manifest in (55), in which $\Delta S > 0$, whereas the probe calculation yields $\Delta S < 0$. Nevertheless, in both calculations, the phase transition is detectable. A final comment is on the discontinuity of the first derivative of the free energy across the transition. This is obtained by computing $(\partial S)/(\partial d) \sim d^{-1/2} \to \infty$[9] as $d \to 0$. Hence the phase transition is accompanied by a generally divergent discontinuity, except in a fine-tuned limit $T \to 0$, in which it can become a finite quantity. Alternatively, we can consider a derivative with respect to $\mu$, which will remain finite and the corresponding phase transition will be associated with a finite discontinuity of the derivative. Finally note that, as $T \to 1$, the free energy and all other associated physical quantities diverge. This is a singular limit, in which the EOW brane coincides with the conformal boundary of AdS and cuts-off the entire geometry.

Let us now briefly discuss the dual CFT picture. The insertion of an EOW-brane in the bulk amounts to introducing a boundary in the dual CFT, following the proposals in [84, 85]. These boundaries preserve conformal symmetries and the corresponding boundary states are obtained by solving $\left(L_p - \bar{L}_{-p}\right)|B\rangle = 0$, where $\{L_p, \bar{L}_q\}$ are the holomorphic and anti-holomorphic copies of Virasoro generators. A general boundary state $|B\rangle$ can be constructed from a linear combination of the so-called Ishibashi states [86]. As Fig. 2 demonstrates, the CFT is defined on $x \in [-\infty, \infty]$ and $\tau \in [0, \infty]$. The corresponding boundary state can be labelled by an index $|B_\alpha\rangle$, which is encoded in the tension of the brane. Subsequently, for a CFT defined on a cylinder, the Euclidean on-shell action is related to the disc partition function for the BCFT, given by $\langle 0 | B_\alpha \rangle \equiv g_\alpha$. Note, however, that we started with a Poincaré AdS$_3$ geometry and therefore the Holographic on-shell action is not simply related to the disc partition function. Instead, it computes the CFT partition function defined on the half-plane in $\tau$. Finally, a note of caution: Recall that the action of the bulk Hamiltonian as well as the equations for the tangent curves are obtained starting from a Poincaré AdS bulk. One can also begin with a bulk dual of a BCFT and subsequently analyze the bulk Hamiltonian as well as the tangent curves accordingly. This description contains an EOW-brane to begin with and it will be interesting to analyze this case further. We do not, however, expect any qualitative difference in the physical picture.

### 3.3.2 Two EOW branes

Let us now consider two such EOW-branes. The corresponding action is given by

$$S_{\text{full}} = S_{\text{gravity}} + S_{\text{brane}} + S_{\text{corner}}, \tag{56}$$

$$S_{\text{gravity}} = -\frac{1}{2\kappa^2} \int_{\mathcal{M}} d^3x \sqrt{g}\,(R - 2\Lambda) - \frac{1}{\kappa^2} \int_{\partial \mathcal{M}} d^2y \sqrt{h} K, \tag{57}$$

$$S_{\text{brane}} = \sum_{i=1,2} -\frac{1}{\kappa^2} \int_{\Sigma_i} d^2\sigma \sqrt{\gamma}\,(K - T_i), \tag{58}$$

$$S_{\text{corner}} = -\frac{1}{\kappa^2} \int_{\mathcal{C}} d\xi \sqrt{h_{\mathcal{C}}}\,\Theta_{\Sigma, \partial \mathcal{M}}, \quad \mathcal{C} = \left(\Sigma_1 \cap \partial \mathcal{M}\right) \cup \left(\Sigma_2 \cap \partial \mathcal{M}\right) \cup \left(\Sigma_1 \cap \Sigma_2\right). \tag{59}$$

Here $\Sigma_i$, $i = 1, 2$ denote the two EOW branes and $T_i$ are the corresponding tensions. The equations of motion are still given by (40) and (41) and the corresponding brane solutions are $T_1 = -\cos\phi_1$ and $T_2 = -\cos\phi_2$. Now these two EOW-branes may intersect in the bulk, in which case there is a non-trivial finite contribution to the free energy coming from the

---

[9]Recall that $\mu = \beta \sqrt{|d|}$.

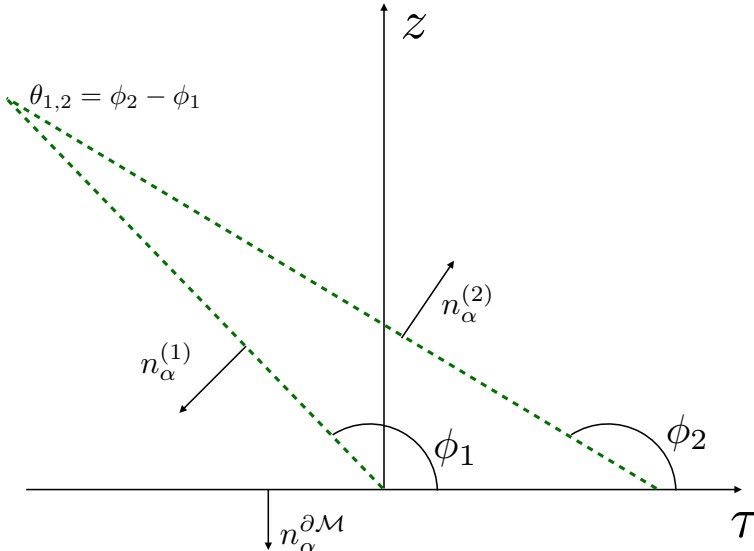

Figure 3: A pictorial representation of two intersecting EOW-branes. The outward normal to $\Sigma_{1,2}$ are denoted by $n_\alpha^{1,2}$ and the outward normal to the conformal boundary of AdS is denoted by $n_\alpha^{\partial\mathcal{M}}$. The two branes intersect at an angle $\theta_{1,2} = \phi_2 - \phi_1$ in the bulk. The gravitational theory is defined within the triangle, including its sides and corners.

intersection point of the two branes. However, if the two branes remain non-intersecting, then there is no such contribution and the result remains the same as above.[10]

The intersecting case is shown in Fig. 3. The analyses proceeds as above with one important addition. Since the outward normals at $\Sigma_1$ and $\Sigma_2$ satisfy: $g^{\alpha\beta} n_\alpha^{(1)} n_\beta^{(2)} < 0$, we choose:

$$n_\alpha^{(1)} = \frac{1}{\sin\phi_1}(0,0,1), \qquad n_\alpha^{(2)} = -\frac{1}{\sin\phi_2}(0,0,1), \tag{60}$$

in both heating and non-heating phases. Here, we have already used the solution for the EOW-brane $\phi'_{1,2} = 0$. From Fig. 3, the branes are intersecting if $\theta_{1,2} = \phi_2 - \phi_1 > 0$;[11] and there is no intersection if $\theta_{1,2} \leq 0$. To proceed further, it convenient to describe these branes in the Poincaré patch:

$$\frac{z}{\tau - \tau_1} = \tan\phi_1, \qquad \frac{z}{\tau - \tau_2} = \tan\phi_2, \tag{61}$$

where $\tau_{1,2}$ are the points at which the EOW-branes $\Sigma_{1,2}$ intersect the conformal boundary of AdS. Their mutual intersection point is given by

$$\tau_* = \frac{\tau_2 \tan\phi_2 - \tau_1 \tan\phi_1}{\tan\phi_1 - \tan\phi_2}, \qquad z_* = \tan\phi_1 \tan\phi_2 \frac{\tau_1 - \tau_2}{\tan\phi_1 - \tan\phi_2}, \tag{62}$$

It is easy to evaluate the pure-gravity part of the on-shell action in both phases. This yields:

$$S_{\text{gravity}} = \frac{1}{\kappa^2}(\tau_1 - \tau_2)\frac{1}{z_*^2}\int dx + \frac{C_1}{\epsilon^2}, \tag{63}$$

---

[10]Recently, similar EOW-branes have been explored in the literature with a different physical motivation. See *e.g.* [81–83, 87] for a representative of such works.

[11]This, in turn, implies that $\tau_1 < \tau_2$.

which consists of either a divergent piece or a finite term that is universal in both phases. Thus, this will not be relevant in the free energy differences. Likewise, the intersection terms $S_{\text{corner}}$ are universal in both phases, except for the contribution coming from the mutual intersection point of the two branes. To proceed further, we now need to fix the ranges of coordinates corresponding to the region enclosed by the branes and the conformal boundary of AdS (see *e.g* Fig. 3).

The domain of interest is defined by the Poincaré coordinates $x \in [-\infty, \infty]$, $\tau \in (\tau_*, \tau_2)$ and $z \in (0, z_*)$. In the non-heating phase, recall that:

$$\tau = -\frac{\sqrt{|d|}}{2} u \cos \phi_{1,2}, \quad z = -\frac{\sqrt{|d|}}{2} u \sin \phi_{1,2}, \quad u = \frac{1}{i}\left[\coth\left(\mu(s+i\theta)\right) - \coth\left(\mu(s-i\theta)\right)\right]. \quad (64)$$

It is clear that $z_* = z_*(s, \theta)$, and therefore the corresponding ranges of the coordinates $\{s, \theta\}$ are mutually dependent. For example, setting $s = 0$,[12] the corresponding coordinate ranges are given by

$$\theta \in \left[\theta_*, \frac{\pi}{2\mu}\right], \qquad \theta_* = \frac{1}{\mu} \cot^{-1}\left(\frac{z_*}{\sqrt{|d|}\sin \phi_{1,2}}\right). \quad (65)$$

Similarly, in the heating phase, one obtains the following range:

$$\theta \in \left[\theta_*, 0\right], \qquad \theta_* = \frac{1}{\mu} \tanh^{-1}\left(\frac{z_*}{\sqrt{|d|}\sin \phi_{1,2}}\right), \quad (66)$$

where we implicitly assume that the intersection point $z_*$ remains within the corresponding patch. Note that, the ranges in (65) and (66) both depend on $z_*$, given a brane angle $\phi_{1,2}$. In general, therefore, $\theta_* = \theta_*(s)$. Thus, the region bounded by the EOW-branes becomes explicitly dynamical. Correspondingly, the on-shell action also depends explicitly on $s$. For simplicity, we will be working with a *free energy density* defined at the $s = 0$ slice, using the above ranges.

In the non-heating phase, the finite contribution from the brane and the corner part evaluates to:

$$S_{\text{brane}} = \frac{2\mu}{\kappa^2}\left[\frac{\cos \phi_1}{\sin^2 \phi_1}\cot\left(2\cot^{-1}\left(\frac{z_*}{\sqrt{|d|}\sin \phi_1}\right)\right) + \frac{\cos \phi_2}{\sin^2 \phi_2}\cot\left(2\cot^{-1}\left(\frac{z_*}{\sqrt{|d|}\sin \phi_2}\right)\right)\right],$$

$$S_{\text{corner}} = -\frac{2\mu}{\kappa^2}\left(\phi_2 - \phi_1\right)\csc\left(2\cot^{-1}\left(\frac{z_*}{\sqrt{|d|}\sin \phi_1}\right)\right). \quad (67)$$

In the heating phase, the corresponding finite contributions are:

$$S_{\text{brane}} = \frac{2\mu}{\kappa^2}\left[\frac{\cos \phi_1}{\sin^2 \phi_1}\coth\left(2\tanh^{-1}\left(\frac{z_*}{\sqrt{|d|}\sin \phi_1}\right)\right)\right.$$

$$\left. + \frac{\cos \phi_2}{\sin^2 \phi_2}\coth\left(2\tanh^{-1}\left(\frac{z_*}{\sqrt{|d|}\sin \phi_2}\right)\right)\right],$$

$$S_{\text{corner}} = -\frac{2\mu}{\kappa^2}\left(\phi_2 - \phi_1\right)\left(\sinh\left(2\tanh^{-1}\left(\frac{z_*}{\sqrt{|d|}\sin \phi_1}\right)\right)\right)^{-1}. \quad (68)$$

---

[12]Recall that this choice does not affect the stroboscopic time $n$ to be a suitably large integer. This is simply because there is always a shift freedom between these two coordinates.

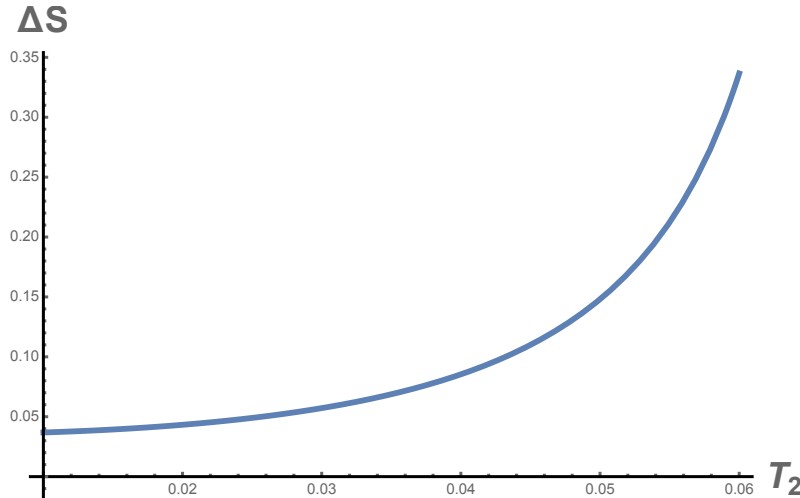

Figure 4: A representative behaviour of the free energy difference $\Delta S = S_{\text{heating}} - S_{\text{non-heating}}$, for fixed values of $T_1 = 0.001$, $|d| = 2$, $\tau_1 - \tau_2 = -0.1$, within a specified range of values of $T_2$ which are shown in the figure. This plot shows a monotonically increasing and a positive $\Delta S$ in this range of $T_2$, with $\Delta S \to \infty$ as $T_2 \to T_2^c \approx 0.0715$. At $T_2 = T_2^c$, there is an infinite jump. Thus, the non-heating phase, in this branch, has a lower free energy.

The free energy difference is now given by taking the difference between (68) and (67).

It is straightforward to check that in the special case when $T_2 = 0$ and $T_1 = T$, we get back the same answer as in (55). In the special case when $T_1 = T_2 = T$, we also get back the same qualitative physics, since the free energies are enhanced by a factor of two, keeping the sign and the behaviour of the difference the same. The general behaviour is richer. A representative feature is shown in Figs. 4 and 5. For a given $T_1$, the free energy difference has two distinct signatures in two regimes of $T_2$. These two regimes are demarcated by the point at $z_* = \sqrt{|d|} \sin \phi_1$, which yields:

$$\phi_2 = \arctan \left( \frac{\sqrt{|d|} \sin \phi_1}{\tau_1 - \tau_2 + \sqrt{|d|} \cos \phi_1} \right). \tag{69}$$

At this location $\Delta S \to \pm \infty$, as $\phi_2$ approaches the above value from above or from below.

Let us now discuss the dual CFT perspective. The presence of two boundaries in the CFT has two different interpretations for the corresponding BCFT. In the so-called open string channel, one considers an open string with two end points at the two boundaries. Alternatively, one can adopt a closed-string channel description, in which case a closed string state evolves from an initial state to a final state. Consider the Euclidean path integral, denoted by $Z_{ab}$, of a CFT on a cylinder with circumference $\tau_\beta$ and vertical width $\tau_w$, with boundary conditions $a$ and $b$ at the two ends. See Fig. 6 for a pictorial representation. As before, we add a note of caution: One can alternatively begin with a bulk geometry with the EOW-branes already inserted and explore the corresponding patches by analyzing the bulk Hamiltonian and the corresponding tangent curves. We expect the key qualitative aspect to remain unchanged, however, it is an interesting scenario to explore in detail.

In the open-string channel, $Z_{ab}$ can be thought of as a thermal partition function for a system defined within an interval of width $\tau_w$ with boundary conditions $a$ and $b$ at the two end points. Thus, $Z_{ab} = \text{Tr}(e^{-\tau_\beta H_{\text{open}}})$. In the closed string channel, this becomes a transition amplitude between two boundary states, $|a\rangle$ and $|b\rangle$, in a system which is defined on a circle of circumference $\tau_\beta$. Thus, $Z_{ab} = \langle a | e^{-\tau_w H_{\text{closed}}} | b \rangle$. Note that this geometry is characterized

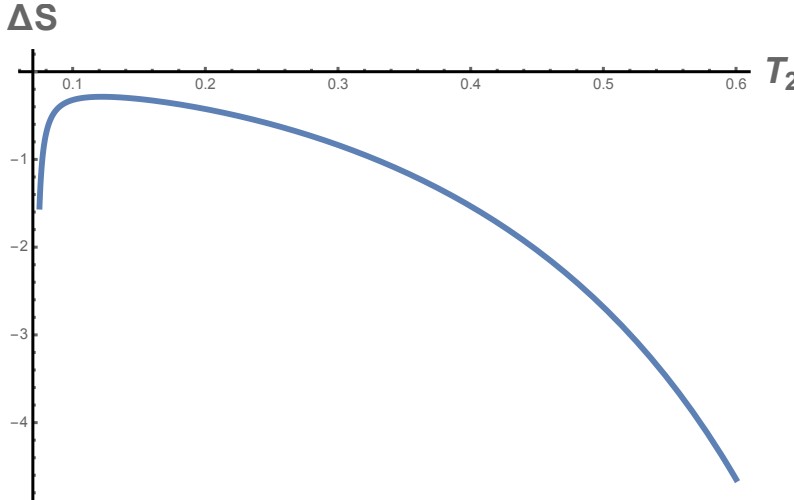

Figure 5: A representative behaviour of the free energy difference $\Delta S = S_{\text{heating}} - S_{\text{non-heating}}$, for fixed values of $T_1 = 0.001$, $|d| = 2$, $\tau_1 - \tau_2 = -0.1$, within a specified range of values of $T_2$ which are shown in the figure. This plot shows a negative $\Delta S$ in this range of $T_2 > T_2^c \approx 0.0715$. Thus, the heating phase has a lower free energy in this branch.

by the dimensionless ratio $\tau_w/\tau_\beta$, up to its conformal class. It can be shown that in the limit $\tau_w/\tau_\beta \to \infty$, the Euclidean path integral is given by: $Z_{ab} = g_a g_b e^{\frac{\pi c \tau_w}{6\tau_\beta}}$, where $c$ is the central charge of the CFT and $g_{a,b} = \langle a, b | 0 \rangle$. These $g_{a,b}$ are ground state degeneracies. In the limit, $\tau_w/\tau_\beta \to \infty$, the Holographic on-shell action is precisely related to these ground state degeneracies. Note, however, that our bulk dual is based on the Poincaré AdS$_3$ geometry and therefore the CFT is defined on a decompactified circle: $\tau_\beta \to \infty$. In this limit, the bulk on-shell action still computes the CFT partition function $Z_{ab}$, and this receives contribution from ground state as well as excited states. To precisely connect with boundary entropy of the BCFT, one should begin with a global AdS$_3$ geometry and subsequently carry out the analyses above. It is an interesting aspect, which we leave for a future work.

A final note is about the types of boundary conditions. With two EOW-branes, *i.e.* with two boundaries one can define a boundary condition changing operator. These operators are formally defined as the primary operators with the smallest dimension, in the spectrum of open string channel with two non-identical boundary conditions at the two end points. This is non-trivial when $a \neq b$, which corresponds two EOW-branes with two different tensions $T_1 \neq T_2$. As we have demonstrated above, this has a rich structure associated with the phase transition.

# 4 Boundary correlation functions from the bulk geometry

In this section, we compute two-point and four-point correlation functions in the bulk in all three phases. We then compare them with the known results in a large $c$ CFT [61, 65]. This will provide a self-consistency check on our geometric description.

## 4.1 Two-point correlation functions

We will compute the two-point function using the geodesic approximation [66] [67], wherein the two point function is approximated by the exponential of the geodesic distance between the

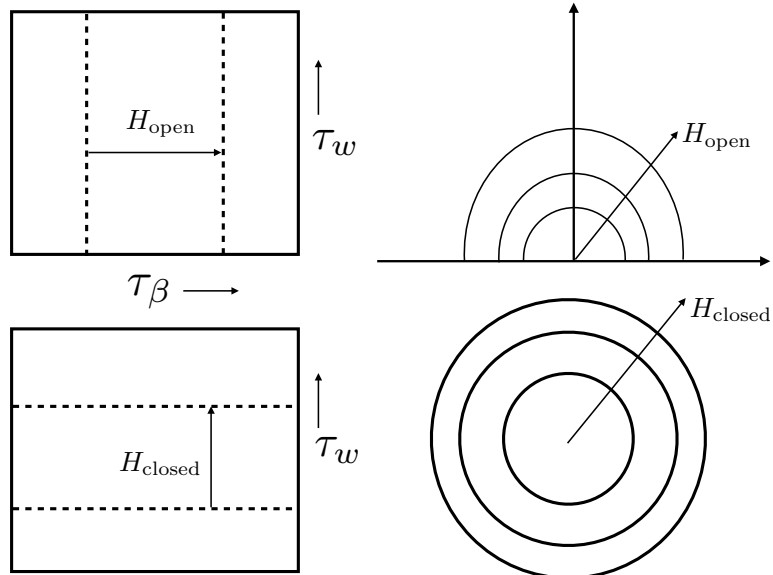

Figure 6: A pictorial representation of the BCFT partition function, which is defined on a rectangular region of horizontal length $\tau_\beta$ and a vertical length of $\tau_w$. Let us assume that $\tau_\beta$ is periodic. The picture above corresponds to the open string channel, in which an open string has end points at the two horizontal lines, separated by $\tau_w$. The corresponding CFT can be defined on the upper half plane, which is shown on the right. The picture below corresponds to the closed string channel, in which a closed string of circumference $\tau_\beta$ propagates from an initial state to a final state. The corresponding CFT is defined on the entire plane, with different states inserted on circles of different radii.

two boundary operators. To this end, we first express the geodesic in terms of the embedding coordinates $(T_1, T_2, X_1, X_2)$ of $AdS_3$.

The geodesic distance ($D$) between two points whose embedding coordinates are $(T_1, T_2, X_1, X_2)$ and $(T_1', T_2', X_1', X_2')$, is given by [67][13]

$$\cosh D = T_1 T_1' + T_2 T_2' - X_1 X_1' - X_2 X_2'. \tag{70}$$

We then express the answer in terms of the bulk coordinates by using the explicit map between the bulk coordinates in which the bulk metric is written for each of the phases and the embedding coordinates. We do this for each phase separately and after regulating the divergence in the geodesic distance, we find an exact match with the boundary two-point function.

### 4.1.1 2-point correlation function in the heating phase

We start with the metric (21) for the heating phase ($d < 0$) and consider the following coordinate changes:

$$r = \coth(2\mu\theta), \quad \text{and} \quad t = 2\mu s, \tag{71}$$

to rewrite (21) as:

$$ds^2 = \frac{d\phi^2}{\sin^2 \phi} + \frac{1}{\sin^2 \phi}\left(\frac{dr^2}{r^2 - 1} - (r^2 - 1)dt^2\right). \tag{72}$$

---

[13]We have set the AdS radius $l = 1$.

The embedding coordinates corresponding to (72) are given by:

$$
\begin{aligned}
T_1 &= \sqrt{r^2-1}\,\sinh t\,\csc\phi\,,\\
T_2 &= r\,\csc\phi\,,\\
X_1 &= \cot\phi\,,\\
X_2 &= \sqrt{r^2-1}\,\cosh t\,\csc\phi\,.
\end{aligned}
\tag{73}
$$

We now compute the correlators of two boundary operators $V$[14] which are located at $(t_1, r_1, \phi = 0)$ and $(t_2, r_2, \phi = 0)$.

In this set-up, the geodesic length in (70) turns out to be:

$$
\cosh D = \lambda^2\left(r_1 r_2 - 1 - \sqrt{(r_1^2-1)(r_2^2-1)}\,\cosh(t_1-t_2)\right).
\tag{74}
$$

Here $\lambda \equiv \csc\phi$. For the boundary points, this is actually divergent since the boundary points are at $\phi = 0$. We therefore regulate it by taking $\lambda$ large but not infinite and then removing the divergent term to obtain the regularized geodesic distance. In this limit, the expression simplifies to:

$$
D \sim \log\left[\left(2\lambda\sqrt{r_1^2-1}\right)\left(2\lambda\sqrt{r_2^2-1}\right)\frac{r_1 r_2 - 1 - \sqrt{(r_1^2-1)(r_2^2-1)}\,\cosh(t_1-t_2)}{2\sqrt{(r_1^2-1)(r_2^2-1)}}\right].
\tag{75}
$$

Using (74), and removing the regulator term $2\lambda\sqrt{(r^2-1)}$, the two point correlator becomes:

$$
\langle VV\rangle \sim e^{-mD} = \left(\frac{2\sqrt{r_1^2-1}\sqrt{r_2^2-1}}{r_1 r_2 - 1 - \sqrt{r_1^2-1}\sqrt{r_2^2-1}\,\cosh(t_1-t_2)}\right)^m.
\tag{76}
$$

This matches exactly with the boundary computation as we now show.

**Boundary computation of the two point function**:

The boundary theory lives at $\phi = 0$. From the curve equations in (14) in the heating phase, at $\phi = 0$ we get:

$$
x + i\tau = z = \sqrt{d}\,\tan\mu(n+i\theta),
\tag{77}
$$

$$
x - i\tau = \bar{z} = \sqrt{d}\,\tan\mu(n-i\theta).
\tag{78}
$$

Here we have identified $n$ with the boundary time $s$.

If we define $\omega = \mu(n+i\theta)$, then the above equations become:

$$
z = \sqrt{d}\,\tan\omega\,,
\tag{79}
$$

$$
\bar{z} = \sqrt{d}\,\tan\bar{\omega}\,.
\tag{80}
$$

Using (79) and (80) the two point function can be written as:

$$
\begin{aligned}
\langle\Phi(\omega_1,\bar{\omega}_1)\Phi(\omega_2,\bar{\omega}_2)\rangle &= \left(\frac{\partial\omega_1}{\partial z_1}\right)^{-h}\left(\frac{\partial\omega_2}{\partial z_2}\right)^{-h}\left(\frac{\partial\bar{\omega}_1}{\partial\bar{z}_1}\right)^{-h}\left(\frac{\partial\bar{\omega}_2}{\partial\bar{z}_2}\right)^{-h}\frac{1}{(z_1-z_2)^{2h}(\bar{z}_1-\bar{z}_2)^{2h}}\\
&= \frac{2^{2h}}{\left(\cos 2\mu(n_1-n_2) - \cosh 2\mu(\theta_1-\theta_2)\right)^{2h}}.
\end{aligned}
$$

---

[14]Here, the operators in consideration are heavy operators of mass $m$.

We analytically continue $n \to in$ to get the Lorentzian correlator as:

$$\langle \Phi(\omega_1, \bar{\omega}_1) \Phi(\omega_2, \bar{\omega}_2) \rangle = \frac{2^{2h}}{\left( \cosh 2\mu(n_1 - n_2) - \cosh 2\mu(\theta_1 - \theta_2) \right)^{2h}} \, . \tag{81}$$

Re-defining $t = 2\mu n$ and $\coth 2\mu\theta = r$,

$$\cosh 2\mu(\theta_1 - \theta_2) = \frac{r_1 - r_2}{\sqrt{(r_1^2 - 1)(r_2^2 - 1)}} \, . \tag{82}$$

Substituting (82) in (81), we get:

$$\langle \Phi(\omega_1, \bar{\omega}_1) \Phi(\omega_2, \bar{\omega}_2) \rangle = \left[ \frac{2\sqrt{(r_1^2 - 1)(r_2^2 - 1)}}{(\cosh(t_1 - t_2)\sqrt{(r_1^2 - 1)(r_2^2 - 1)} - r_1 r_2 + 1)} \right]^{2h} \, . \tag{83}$$

For heavy operators identifying $m \sim 2h_v$, we get an exact match with the bulk answer given in (76).

### 4.1.2  2-point correlation function in the non-heating phase

We can rewrite the metric (19) corresponding to the non-heating phase ($d > 0$) as

$$ds^2 = \frac{d\phi^2}{\sin^2 \phi} + \frac{1}{\sin^2 \phi} \left( \frac{dr^2}{r^2 + 1} - (r^2 + 1) dt^2 \right), \tag{84}$$

by considering the following coordinate change

$$r = \cot(2\mu\theta), \quad \text{and} \quad t = 2\mu s \, . \tag{85}$$

The embedding coordinates are given by:

$$\begin{aligned}
T_1 &= \sqrt{r^2 + 1} \sin t \csc \phi \, , \\
T_2 &= \sqrt{r^2 + 1} \cos t \csc \phi \, , \\
X_1 &= \cot \phi \, , \\
X_2 &= r \csc \phi \, .
\end{aligned} \tag{86}$$

The geodesic length, given by (70), is:

$$\cosh D = \lambda^2 \left( \sqrt{(r_1^2 + 1)(r_2^2 + 1)} \cos(t_1 - t_2) - 1 - r_1 r_2 \right) . \tag{87}$$

Once again, the distance is divergent and we need to regulate it. Hence, as in the heating phase case, we obtain

$$D \sim \log \left[ \left( 2\lambda \sqrt{r_1^2 + 1} \right) \left( 2\lambda \sqrt{r_2^2 + 1} \right) \frac{\sqrt{(r_1^2 + 1)(r_2^2 + 1)} \cos(t_1 - t_2) - r_1 r_2 - 1}{2\sqrt{(r_1^2 + 1)(r_2^2 + 1)}} \right] . \tag{88}$$

Using (88), the two point correlator is obtained to be:

$$\langle VV \rangle \sim e^{-mD} = \left( \frac{2\sqrt{r_1^2 + 1}\sqrt{r_2^2 + 1}}{\sqrt{r_1^2 + 1}\sqrt{r_2^2 + 1} \cos(t_1 - t_2) - r_1 r_2 - 1} \right)^m . \tag{89}$$

In the above, the geodesic distance has been regulated with a regulator $2\lambda\sqrt{(r^2+1)}$.

**The boundary computation**:

At $\phi = 0$, the curve equations in (10) gives:

$$x + i\tau = z = -\sqrt{d}\coth\mu(n + i\theta) = -\sqrt{d}\coth\omega\,, \tag{90}$$

$$x - i\tau = \bar{z} = -\sqrt{d}\coth\mu(n - i\theta) = -\sqrt{d}\coth\bar{\omega}\,. \tag{91}$$

Using the above equations the two point function in this case can be written as:

$$\langle\Phi(\omega_1,\bar{\omega}_1)\Phi(\omega_2,\bar{\omega}_2)\rangle = \left(\frac{\partial\omega_1}{\partial z_1}\right)^{-h}\left(\frac{\partial\omega_2}{\partial z_2}\right)^{-h}\left(\frac{\partial\bar{\omega}_1}{\partial\bar{z}_1}\right)^{-h}\left(\frac{\partial\bar{\omega}_2}{\partial\bar{z}_2}\right)^{-h}\frac{1}{(z_1-z_2)^{2h}(\bar{z}_1-\bar{z}_2)^{2h}}$$

$$= \frac{2^{2h}}{(\cos 2\mu(n_1-n_2)-\cos 2\mu(\theta_1-\theta_2))^{2h}}\,.$$

The Lorenzian correlator for non-heating case after analytic continuation ($n \to in$) and coordinate re-definition ($t = 2\mu n$, $r = \cot 2\mu\theta$) is:

$$\langle\Phi(\omega_1,\bar{\omega}_1)\Phi(\omega_2,\bar{\omega}_2)\rangle = \left[\frac{2\sqrt{(r_1^2+1)(r_2^2+1)}}{(\cos(t_1-t_2)\sqrt{(r_1^2+1)(r_2^2+1)}-r_1 r_2-1)}\right]^{2h}\,. \tag{92}$$

Which matches with the expression derived from the bulk in equation (89)

### 4.1.3   2-point correlation function in the phase boundary

The embedding coordinates for this case can be written down as:

$$T_1 = \frac{1}{2r}(1 + r^2(1 - t^2))\csc\phi\,,$$
$$T_2 = t\, r\csc\phi\,,$$
$$X_1 = \cot\phi\,,$$
$$X_2 = \frac{1}{2r}(1 - r^2(1 + t^2))\csc\phi\,. \tag{93}$$

In this case the corresponding metric in (18), after the coordinate change $r = \frac{1}{\theta}$ and $t = s$, can be rewritten as,

$$ds^2 = \frac{d\phi^2}{\sin^2\phi} + \frac{1}{\sin^2\phi}\left(\frac{dr^2}{r^2} - r^2 dt^2\right). \tag{94}$$

The corresponding geodesic length is:

$$\cosh D = \frac{\lambda^2}{2r_1 r_2}\left((r_1 - r_2)^2 - (r_1 r_2)^2(t_1 - t_2)^2\right). \tag{95}$$

Similar to the previous cases, using (95), we find the regulated geodesic length, with regulator ($r\lambda = \frac{\csc\phi}{\theta}$) and then the two point correlator is obtained to be:

$$\langle VV\rangle \sim e^{-mD} = \left(\frac{r_1^2 r_2^2}{(r_1 - r_2)^2 - r_1^2 r_2^2(t_1 - t_2)^2}\right)^m\,. \tag{96}$$

**The boundary computation**:

As before, we start with (17) at $\phi = 0$ and find the two point correlator after analytically continuing $n \to in$ and suitably redefining coordinates $t = n, r = \frac{1}{\theta}$ to be:

$$\langle \Phi(\omega_1, \bar{\omega}_1) \Phi(\omega_2, \bar{\omega}_2) \rangle = \left[ \frac{(r_1^2 r_2^2)}{(r_1 - r_2)^2 - r_1^2 r_2^2 (t_1 - t_2)^2} \right]^{2h}. \tag{97}$$

Again this matches exactly with (96).

## 4.2 4-point out of time order correlators from the bulk

In this section we compute a 4-point OTOC in the bulk geometry following the work of [66,67]. The idea, as argued in [67], is that the four point OTOC in the bulk can be thought of as a two point function in a perturbed shock wave geometry created by one of the operators. In this section, we will set up the computation in the heating phase geometry. We will show the emergence of a exponential temporal behaviour at late times with a Lyapunov exponent which will exactly match with the boundary value obtained in [65]. We then end the section by pointing out the crucial difference with the other two phases, which will lead to a non-exponential temporal behaviour in those cases.

### 4.2.1 The shock wave profile

We begin with a derivation of the shock-wave profile following the seminal work of [88]. We start with the form of metric given in 72:

$$ds^2 = \frac{d\phi^2}{\sin^2 \phi} + \frac{1}{\sin^2 \phi} \left( \frac{dr^2}{r^2 - 1} - (r^2 - 1) dt^2 \right). \tag{98}$$

In terms of Kruskal coordinates, this takes the form:

$$ds^2 = \frac{d\phi^2}{\sin^2 \phi} + \frac{1}{\sin^2 \phi} \left( \frac{-4}{(1 + uv)^2} du dv \right), \tag{99}$$

where, $u = -e^{-\tilde{u}}$, $v = e^{\tilde{v}}$ with $\tilde{u} = t - r_*$, $\tilde{v} = t + r_*$ and $r_* = \frac{1}{2} \ln \frac{|r-1|}{r+1}$.

The metric (72) has a horizon at $r = 1$ or $uv = -1$. This will then represent a two-sided black-hole geometry in extended Kruskal coordinates. The boundary theory lives at $\phi = 0$ hyper-surface. The above metric (99) is of the following form [89]:

$$ds^2 = 2A(u, v) h(\phi) du dv + h(\phi) d\phi^2, \tag{100}$$

where $A(u, v) = \frac{-4}{(1+uv)^2}$ and $h(\phi) = \frac{1}{\sin^2 \phi}$. Consider a scenario where a massless particle at $u = 0$ moves along the $v$-direction in the background metric (100), along a constant ($\phi = a$) which back-reacts and results in a shock wave geometry. Following [88], our ansatz for the form of the shock wave geometry is:

$$ds^2 = 2A(\tilde{u}, \tilde{v}) h(\tilde{\phi}) d\tilde{u} d\tilde{v} - 2A(\tilde{u}, \tilde{v}) h(\tilde{\phi}) \eta \delta(\tilde{u}) d\tilde{u}^2 + h(\tilde{\phi}) d\tilde{\phi}^2. \tag{101}$$

This shock wave geometry in (101) is described by (100) for both $u > 0$ and $u < 0$ with the effect of the shock wave being that the $v$ coordinate for $u > 0$ is shifted to $v + \eta(\phi)$. In (101), $\tilde{v} = v + \eta(\phi) \theta(u)$, $\tilde{u} = u$ and $\tilde{\phi} = \phi$. Our main objective is to determine $\eta(\phi)$ that determines the shock wave profile.[15]

---

[15]Let us note that due to the presence of an overall conformal factor $h(\phi)$, the metric in (100) is slightly different from the form of the metric considered in [88] and [89]. Therefore, we expect the conditions (eg. see Eq. 2.10 of [89]) on metric components and the equation satisfied by the shock wave profile in our case would be different.

The core idea of this calculation is based on the fact that the ansatz metric (101) solves the Einstein equation with appropriate source terms. These source terms are given by the sum of stress tensor of the unperturbed geometry and the stress tensor of the moving particle ($T^p$) with momentum $p$. Here,

$$T^p = T^p_{\tilde{u}\tilde{u}}d\tilde{u}^2 = -4p\, A^2\, h^2(\tilde{\phi})\, \delta(\tilde{u})\, d\tilde{u}^2\,.$$

Subsequently, comparing the coefficients of $\delta(\tilde{u})$ on both sides of the Einstein equation, we get the following conditions:

$$\text{At } \tilde{u} = 0\,, \quad A_{,\tilde{v}} = 0\,, \quad A_{,\tilde{v}\tilde{v}} = 0\,,$$
$$\eta''(\phi) + \frac{h'(\phi)}{2h(\phi)}\eta'(\phi) = 32\pi p\, A\, h^2\, \delta(\phi - a)\,.$$

In our case, this takes the following form:

$$\eta''(\phi) - \cot\phi\ \eta'(\phi) = -\frac{c'\sin a}{\sin^4\phi}\,\delta(\phi - a)\,, \tag{102}$$

where, $c' = 32\pi\, p$. The solution to above equation is:

$$\eta(\phi) = c_2 - c_1\cos\phi + c'\csc^4 a\left[(\cos\phi - \cos a)\Theta(\phi - a)\right]\,. \tag{103}$$

To proceed further, we need to impose boundary conditions to fix the constant $c_1$ and $c_2$. We impose the boundary condition that the shock wave is entirely in the bulk and has no component along the boundary, i.e.: $\eta(\phi) = 0$ at $\phi = 0, \pi$.

This completely determines the profile function, which takes the following form.[16]

$$\eta(\phi) = \frac{c'\csc^4 a}{2}\left[(1 - \cos\phi)(1 + \cos a) + 2(\cos\phi - \cos a)\Theta(\phi - a)\right]\,. \tag{104}$$

### 4.2.2  4-point OTOC in the heating phase geometry

Let us now compute OTOC of two scalar operators ($V$ and $W$) in the bulk. As mentioned earlier, this reduces to the computation of a two point function in a perturbed shock wave geometry [67]. Therefore, we have to compute $_W\langle V_L V_R\rangle_W$ similar to 4.1.1 but in the shock wave geometry produced by a particle $W$, where the two boundary operators ($V_L, V_R$) with mass $m$ are considered to be on left and right boundary of the extended geometry. The shock wave in this case is due to back-reaction produced by the large blue shifted proper energy, denoted by $E_w$, of the probe particle $W$. This $W$ particle is released from the boundary in the far past, at a time $t_w$, as measured by a static observer near horizon at time $t = 0$.[17] We can redefine $t = 2\beta\sqrt{d}\,s$ and $r' = 2\beta\sqrt{d}\,r$, the metric (72) becomes a AdS$_2$ blackhole patch of AdS$_3$ with horizon at $r' = 2\sqrt{d}\beta$:

$$ds^2 = \frac{d\phi^2}{\sin^2\phi} + \frac{1}{\sin^2\phi}\left(\frac{dr'^2}{r'^2 - 4d\beta^2} - (r'^2 - 4d\beta^2)ds^2\right)\,. \tag{105}$$

---

[16]The final expression (104) depends quite non-trivially on the specific choice of boundary conditions i.e. $\eta(\phi) = 0$ at $\phi = 0, \pi$. Given the fact that the calculation's final outcome is heavily reliant on this shock wave profile, one may wonder if there is a particular and distinctive way to select the boundary conditions and whether different boundary conditions will correspond to completely different physical cases. It would be nice to explore these questions further.

[17]See 4.2.3 for more detail.

The boosted large energy at time $t_w = 0$ in the above metric is:

$$E_w \sim \frac{E}{4d\beta^2} \sin a e^{2\sqrt{d}\beta s_w}. \tag{106}$$

As we do in the two-point function computations, we start by writing down geodesic lengths in terms of embedding coordinates but this time we write two separate geodesic distances $d_1$ and $d_2$ from a boundary point to some bulk point on both sides of the shockwave geometry. The actual geodesic is then calculated by extremizing the sum of two distances $d_1 + d_2$, with respect to $v$ and $\phi$ so that it meets the shock wave at $v_*$ on $\phi_*$ slice. Here, $d_1$ refers to the geodesic length from the left boundary point $(t_L = 0, r, \phi_0 = 0)$ to some bulk point at $(u = 0, v, \phi)$, while, $d_2$ is the geodesic length from $(\tilde{u} = u = 0, \tilde{v}, \phi)$ to the right boundary point $(t_R = 0, r, \phi_0 = 0)$. The expressions for $d_1, d_2$ in terms of embedding coordinates in (73) are given by

$$\cosh d_1 = \left[ r + e^{-t_L} \sqrt{r^2 - 1}\, v - \cos\phi\cos\phi_0 \right] \csc\phi \csc\phi_0, \tag{107}$$

$$\cosh d_2 = \left[ r + e^{-t_R} \sqrt{r^2 - 1}\, (v + \eta(\phi)) - \cos\phi\cos\phi_0 \right] \csc\phi \csc\phi_0. \tag{108}$$

Recall that for $\phi_0 = 0$, $\csc\phi_0 = \lambda$ diverges and needs a regularization. The final geodesic length is calculated in two steps: First, by extremizing $d_1 + d_2$ in (107) and (108) with respect to $v$ yields: $v_* = -\eta/2$ and the corresponding geodesic length is given by

$$\cosh\frac{\tilde{d}}{2} = \lambda \left( r + \sqrt{r^2 - 1}\frac{\eta(\phi)}{2} - \cos\phi \right) \csc\phi. \tag{109}$$

Extremizing further with respect to $\phi$ yields:

$$\cos\phi_* = \frac{4 + c'\sqrt{r^2 - 1}(\cos a + 1)\csc^4 a}{4r + c'\sqrt{r^2 - 1}(\cos a + 1)\csc^4 a}, \qquad \text{for} \quad \phi < a,$$

$$\cos\phi_* = \frac{4 + c'\sqrt{r^2 - 1}(\cos a - 1)\csc^4 a}{4r + c'\sqrt{r^2 - 1}(\cos a - 1)\csc^4 a}, \qquad \text{for} \quad \phi > a.$$

Substituting $\phi_*$ back in (73), the final geodesic distance turns out to be:

$$d \approx 2\log\left[ 2\lambda\sqrt{r^2 - 1} \right] + \log\left[ 1 + c'\sqrt{\frac{r - 1}{r + 1}}\left( \frac{\cos a + 1}{\sin^4 a} \right) \right], \quad \text{for } a > \frac{\pi}{2}, \tag{110}$$

or,

$$d \approx 2\log\left[ 2\lambda\sqrt{r^2 - 1} \right] + \log\left[ 1 + c'\sqrt{\frac{r + 1}{r - 1}}\left( \frac{1 - \cos a}{\sin^4 a} \right) \right], \quad \text{for } a < \frac{\pi}{2}. \tag{111}$$

After subtracting the divergent contribution from $2\lambda\sqrt{r^2 - 1}$ and using a geodesic approximation, $_W\langle VV \rangle_W \propto e^{-md}$ with regularized geodesic distance $d$ and substituting $c' \sim 32\pi \frac{E}{4d\beta^2} \sin a e^{2\sqrt{d}\beta s_w}$, we find that the final form of OTOC is:

$$\frac{_W\langle V_L V_R \rangle_W}{\langle WW \rangle \langle V_L V_R \rangle} \approx \left( \frac{1}{1 + 32\pi \frac{E}{4d\beta^2}\sqrt{\frac{r\mp 1}{r\pm 1}}\left( \frac{1\pm\cos a}{\sin^3 a} \right)e^{2\sqrt{d}\beta s_w}} \right)^m. \tag{112}$$

From the above expression we get the Lyapunov exponent to be $2\sqrt{d}\beta$. This matches precisely with the Lyapunov exponent obtained from a purely CFT computation in [65]. In that work,

a direct and explicit CFT calculation was carried out for a large $c$ CFT. With a discrete drive, in the heating phase, the four point OTOC in large-c CFT was obtained to be:

$$\mathcal{F} = \left( \frac{1}{1 - \frac{24\pi i h_w e^{4n\theta}}{c\epsilon_{12}\epsilon_{34}A(z_w,z_v)}} \right)^{2h_v}, \tag{113}$$

where, $A(z_w,z_v) = -16\theta^2 \frac{(z_v-1)(z_w+1)}{(z_v+1)(z_w-1)}$. From the above equation, one can extract the Lyapunov exponent to be:

$$\lambda_{\text{L}} = \frac{4\theta}{(T_1+T_2)}. \tag{114}$$

Expressing the above equation in terms of the parameters of the effective hamiltonian, then using (A.11), we get:

$$\lambda_{\text{L}} = \sqrt{\alpha^2 - 4\beta\gamma} = 2\beta\sqrt{d} = \frac{4\theta}{(T_1+T_2)}. \tag{115}$$

This matches with the bulk computation. It would be nice to match the full expression (113) with its bulk counterpart and not just the Lyapunov exponent. The function $A(z_v,z_w)$ is a non-trivial function of the position of two operators $V$ and $W$. The bulk expression derived here is a function of the position of the boundary $V$ operator, however the only information of the $W$ operator which enters is the direction $\phi = a$ along which the particle which creates the shock wave propagates. We have not been able to translate this information into the boundary location of the $W$ operator. However, it is encouraging that the dependence on the position of the $V$ operator is similar in both the expressions. We hope to be able to return to this in the near future.

### 4.2.3 OTOC in non-heating phase and phase transition

Let us repeat the same calculations in the non-heating phase, as well as on the phase boundary. In the heating case, the exponential behaviour in the OTOC was due to the shockwave geometry that results from the large blue-shifted energy $\mathcal{O}(e_w^t)$ of the $W$ particle which is released at a very early time $t_w$. In general, if a particle released from the boundary $r \to \infty$ at an early time is moving along a null trajectory with proper energy $E$, the energy $E_r$ measured on the time slice $t = 0$, is

$$E_r = \frac{E}{\sqrt{g_{00}|_{t=0}}}. \tag{116}$$

We will now investigate the behavior of $E_r$ for the metrics in other phases.

**For non-heating phase**: We start with the metric (19) and follow exact similar procedure as in the previous section to rewrite the metric in terms of $r = \cot(2\mu\theta)$, $t = 2\mu s$ and $\phi$:

$$ds^2 = \frac{d\phi^2}{\sin^2\phi} + \frac{1}{\sin^2\phi}\left( \frac{dr^2}{r^2+1} - (r^2+1)dt^2 \right). \tag{117}$$

The tortoise coordinate $r_*$ in this case is given by $\frac{dr_*}{dr} = \frac{1}{1+r^2}$ and hence, $r_* = \tan^{-1} r$. Therefore, the trajectory of nearly null $W$-particle released from boundary at $t_w$, in terms of the tortoise coordinate $r_*$ at time t is:

$$t - t_w = r_* - \frac{\pi}{2}. \tag{118}$$

Substituting (118) in (116) we see that for metric (84) the energy measured at time $t = 0$ is:

$$E_r = E \sin a \ \sin t_w \,. \tag{119}$$

**On the phase boundary:**

Similarly, the metric (18) can be re-defined in terms of $r = \frac{1}{\theta}$ and $n = t$. The tortoise coordinate for this case is given by $r_* = \int \frac{dr}{r^2} = -\frac{1}{r}$. Then the null trajectory of $W$-particle in this case is:

$$\int_{t_w}^{t} dt = \int_{0}^{r_*} dr \,,$$
$$\Rightarrow t - t_w = r_* \,.$$

Then using this in (116) we find the energy measured at $t = 0$ is:

$$E_r = E \sin a \ t_w \,. \tag{120}$$

Hence we find that the energy measured at time $t = 0$ in non-heating phase (119) and during phase transition (120) respectively show oscillatory and power law dependence on $t_w$. This is consistent with the boundary results [65].

## 5 Discussions

In this article, we constructed a Holographic description of a (gravity + brane) system which is capable of detecting the non-heating to heating phase transition in the dual boundary CFT, which is subject to a periodic drive. While this framework is completely natural and intuitive in this respect, our construction should be viewed as the simplest of the richer possibilities.[18] Subsequently, there are several intriguing aspects for future explorations. We enlist some of them below.

First, note that the periodically driven Hamiltonian is $sl(2,R)$-valued and therefore does not accommodate the possibilities of a large gauge transformation. The general class of Brown-Henneaux diffeomorphisms contain an infinite number of such large gauge transformations, which are dual to a periodically driven Hamiltonian valued in the $sl^{(q)}(2,R)$, for $q > 1$. Conceptually, it is no harder to find the corresponding curves in the bulk which would be generated by the bulk Hamiltonian. Subsequently, the various patches will likely contain a richer class of metrics, including dynamical ones. This generalization will naturally include the possibility of analyzing a non-trivial highest weight state and its corresponding evolution in the bulk geometric description, which are expected to lie within the family of Banados geometries. It will be an interesting question to consider these cases in detail, also in the presence of EOW-branes. For the conceptual richness and the technical involvement, this deserves an independent study which we hope to address in near future.

A much simpler problem is to consider the $sl(2,R)$-valued drive Hamiltonian and work out the corresponding phase patches starting from a global AdS$_3$. In this case, the CFT is defined on a cylinder and the corresponding (gravity + brane) on-shell action corresponds to the boundary entropy of the dual BCFT. This boundary entropy counts the ground-state degeneracy in the BCFT and it will be interesting to understand in detail how this counting detects the phase transition. This further generalizes in the presence of more than one EOW-branes, with different tensions.

---

[18]Note that, the richness of a boundary degree of freedom in dynamical context has been explored also in [90–92] in the probe limit and in [93–95] away from any probe approximation.

Relatedly, we can explore an alternative way of inserting the EOW-branes in the bulk. One can begin with a bulk Hamiltonian, in a geometry where EOW-branes are already inserted and subsequently analyze the tangent curves and the corresponding induced geometries. This is conceptually different from what we have done here. Although we expect the qualitative features to remain the same, especially so since the EOW-branes emerge naturally in the corresponding patches that we have considered here, it will nonetheless be an interesting issue to understand in greater detail.

It is interesting to note the similarities with the framework in [96], in which a path integral optimization in CFT has been realized in terms of a holographic description. The similarities between these are worth exploring further, especially focussing on the potential connection between phase transition detection and path integral optimization as well as the holographic path-integral complexity.

A crucial point of our study is the appearance of $AdS_2$ slicing which plays a pivotal role in distinguishing phases in terms of unequal time correlators as well as provides a natural setting to incorporate EOW brane. In particular, the OTOC computation strongly suggests that the $AdS_2$ physics is responsible for the different temporal growth in different phases. From the boundary perspective it is not at all clear why such $AdS_2$ foliation emerges. For instance, from Eq. (77) and (90) the boundary tangent curve parametrizes a time dependent boundary metric. This time dependence in boundary metric is responsible for the different temporal behavior of the unequal time correlators. However when we lift those boundary metrics to the bulk $AdS_3$ we end up with time independent $AdS_2$ slicing of $AdS_3$. The presence of the EOW-branes in an AdS-background suggests a doubly-Holographic model structure. Such models have recently been intensely explored in connection with the black hole information paradox [97]-[98]. It will be intriguing if there is a clear connection between the physics of the transition with the physics of the information paradox here. We hope to return with a more clear answer in future. Also note that our brane-analyses are explicitly tied to the choice of static gauge for the brane profile. In general, a relaxation of this condition is technically viable and it remains to be seen how crucially the physics of the phase transition depends on this choice.

Relatedly, it will be interesting to construct examples in which the black hole on the brane becomes truly dynamical. This aspect is expected to be visible with a periodic drive with an $sl^{(q)}(2, R)$-valued Hamiltonian. Alternatively, similar dynamical situation could be appeared in a primary state under the $sl(2, R)$ drive. We would like to address some of these issues in future.

Continuing on the point above, the class of time-dependent Hamiltonians that we have considered is certainly not of the most general kind. There are several possible generalizations that may allow sufficient controlled calculations. For example, a potential generalization to higher dimensional cases appear especially intriguing and some preliminary work is underway to explore this possibility.

Finally, since a connection with the doubly-holographic models emerge in the 2d CFT cases, it will be very interesting to further sharpen the precise connection between such models in higher dimensional CFTs. In recent times, doubly-holographic models have been widely used to address salient features of black hole information recovery where one couples the black hole with a non-gravitational bath. Within our driven CFT framework, such a coupling may be natural for various subsequent physics questions, across dimensions.

## Acknowledgments

We would like to thank Tarek Anous, Parthajit Biswas, Pawel Caputa, Ashish Chandra, Diptarka Das, Damian Galante, Dongsheng Ge, Chethan Krishnan, Sinong Liu, Vinay Malvimat, Giuseppe Policastro, Koushik Ray, Sanjit Shashi and Ritam Sinha for useful conversations on related topics. We would also like to thank anonymous referees for several interesting and useful comments on the draft.

SD would like to acknowledge the hospitality of University of Amsterdam and University of Warsaw where part of the work was presented. BE would like to acknowledge the hospitality of the Physics department at IIT Kanpur, where some of the results were presented. We would also like to thank the Organizers of Out of Equilibrium Physics, held at Indian Institute of Technology, Mandi for a stimulating environment where preliminary results related to this project were presented. SD, SP and BR would like to thank the organizers of ST$^4$ 2022, held at Indian Institute of Technology, Indore, where part of the work have been discussed.

**Funding information** SD would like to acknowledge the support provided by the Max Planck Partner Group grant MAXPLA/PHY/2018577. BE is supported by CRG/2021/004539, AK acknowledges support from the Department of Atomic Energy, Govt. of India, Board of Research in Nuclear Sciences (58/14/12/2021-BRNS) and IFCPAR/CEFIPRA 6304-3. The work of SP and BR is supported by a Senior Research Fellowship(SRF) from UGC. KS thanks DST, India for support through project JCB/2021/000030.

## A  Floquet (effective) Hamiltonian of a driven CFT

In this appendix, we explicitly construct the Floquet (Effective) Hamiltonian ($H_{\text{eff}}$) for the two period discrete drive protocol. The form of Floquet Hamiltonian depends on the driving protocol of the CFT. As an example, we compute the Floquet Hamiltonian of a discretely (two-step) driven CFT where, where, the Hamiltonian $H_\theta = \int_0^L T_{00}\left(1 - \tanh(2\theta)\cos(\frac{2\pi x}{L})\right)dx$ in each period switches between $H_0 = H_{\theta=0}, H_1 = H_{\theta\neq0}$ as in Fig. 7 [57].

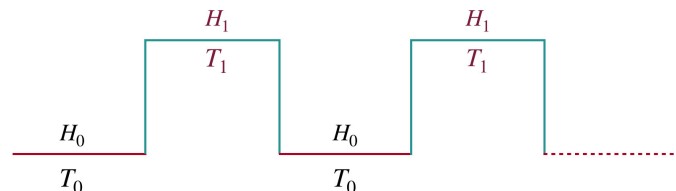

Figure 7: Pictorial representation of discrete drive protocol.

In terms of the modes, the Hamiltonians are given as following:

$$H_0 = \frac{2\pi}{L}\left[L_0 + \bar{L_0}\right] - \frac{\pi c}{12L},$$

and

$$H_1 = \frac{2\pi}{L}\left[L_0 - \tanh(2\theta)\frac{L_1 + L_{-1}}{2}\right] - \frac{\pi c}{12L} + \text{anti-holomorphic part.}$$

At this point, we make an ansatz for the Floquet Hamiltonian, that replicates the same dynamics as the original system. We assume the following as the Floquet Hamiltonian[19]

$$H_{\text{eff}} = [\alpha L_0 + \beta L_1 + \gamma L_{-1}] + \text{anti-holomorphic part(A.H)}.$$

Since $H_0$ and $H_1$ are only made of $L_0, L_{\pm 1}$, the BCH formula guarantees that the Floquet Hamiltonian should be made of only by those global conformal generators. We may able to determine $\alpha, \beta, \gamma$ by demanding that $H_{\text{eff}}$ must satisfy the following condition:

$$e^{-\tau_0 H_0} e^{-\tau_1 H_1}.z = e^{-(\tau_0 + \tau_1)H_{\text{eff}}}.z. \tag{A.1}$$

For discrete drive, the LHS of the above equation gives [55],

$$e^{-\tau_0 H_0} e^{-\tau_1 H_1}.z = \frac{\left[(1-\delta)\cosh 2\theta - (\delta+1)\right]\left(\frac{\delta'}{2\sqrt{\delta\delta'}}\right)z + \frac{(\delta-1)}{\sqrt{2\delta\delta'}}\sinh 2\theta}{\left[(1-\delta)\sinh 2\theta\right]\left(\frac{\delta'}{2\sqrt{\delta\delta'}}\right)z + \frac{1}{\sqrt{2\delta\delta'}}\left[(\delta-1)\cosh 2\theta - (\delta+1)\right]}, \tag{A.2}$$

where $\delta := e^{\frac{2\pi\tau_1}{L\cosh 2\theta}}$ and $\delta' := e^{\frac{2\pi\tau_0}{L}}$. First, we'll compute the action of $H_{\text{eff}}$ on $z$. We will then compare the result with (A.2) to derive the relations between the parameters $\alpha, \beta, \gamma$ and the parameters of the drive. In the $z$ plane, $H_{\text{eff}}$ is

$$H_{\text{eff}} = \int \frac{dz}{2\pi i}(\alpha z + \beta z^2 + \gamma)T(z) + \text{A.H}. \tag{A.3}$$

To simplify (A.3) further, we map it to $\tilde{z}$ plane such that $H_{\text{eff}}$ becomes

$$H_{\text{eff}} = \int \frac{d\tilde{z}}{2\pi i}\tilde{z}T(\tilde{z}) + \text{A.H} = \tilde{L}_0, \tag{A.4}$$

where $\tilde{z}$ and $z$ are related by the following transformation:

$$\tilde{z} = \left[\frac{c'(z-A)}{z-B}\right]^{\frac{1}{\sqrt{(\alpha^2 - 4\beta\gamma)}}}, \tag{A.5}$$

where, $c'$ is a constant, $A = \frac{-\alpha + \sqrt{(\alpha^2 - 4\beta\gamma)}}{2\beta}$ and $B = \frac{-\alpha - \sqrt{(\alpha^2 - 4\beta\gamma)}}{2\beta}$. In the $\tilde{z}$ plane, $H_{eff}$ acts as,

$$e^{-sH_{eff}}\tilde{z} = e^{-s\tilde{L}_0} = e^s\tilde{z}, \tag{A.6}$$

using the above identity and (A.5), we found that

$$z' = e^{-\tau_0 H_0} e^{-\tau_1 H_1}.z = e^{-(\tau_0 + \tau_1)H_{\text{eff}}}.z(\tilde{z}) = z(e^{-(\tau_0+\tau_1)}\tilde{z})$$

$$= \frac{\frac{Am^{\frac{-1}{2}} - Bm^{\frac{1}{2}}}{A-B}z + \frac{AB}{A-B}\left(m^{\frac{1}{2}} - m^{\frac{-1}{2}}\right)}{\left(m^{\frac{-1}{2}} - m^{\frac{1}{2}}\right)\frac{z}{A-B} + \frac{Am^{\frac{-1}{2}} - Bm^{\frac{1}{2}}}{A-B}}, \tag{A.7}$$

---

[19]Here we ignore the c-number part which is irrelevant for our purpose.

here, $m = e^{(\tau_0 + \tau_1)\sqrt{(\alpha^2 - 4\beta\gamma)}}$. After comparing (A.7) with (A.2) we found that,

$$\frac{\beta}{\alpha} = \frac{\delta'(1-\delta)\sinh 2\theta}{(\delta+1)(1-\delta')-(\delta-1)(1+\delta')\cosh 2\theta}, \tag{A.8}$$

$$\frac{\gamma}{\alpha} = \frac{(1-\delta)\sinh 2\theta}{(\delta+1)(1-\delta')-(\delta-1)(1+\delta')\cosh 2\theta}, \tag{A.9}$$

$$m = \left[ \frac{\left( \left( (1+\delta)(1+\delta') + (1-\delta)(1-\delta')\cosh 2\theta \right)^2 - 16\delta\delta' \right)^{\frac{1}{2}}}{4(\delta\delta')^{\frac{1}{2}}} \right.$$

$$\left. + \frac{(1+\delta)(1+\delta') + (1-\delta)(1-\delta')\cosh 2\theta}{4(\delta\delta')^{\frac{1}{2}}} \right]^2. \tag{A.10}$$

In the discrete drive protocol, a simple choice of drive frequencies leads to a heating phase: $i\tau_0 \equiv T_0 = \frac{L}{2}, i\tau_1 \equiv T_1 = \frac{L\cosh(2\theta)}{2}$ such that the su(1,1) transfer matrix takes a simple form: $a_n = d_n = (-1)^n \cosh(2n\theta); b_n = c_n = -(-1)^n \sinh(2n\theta)$ [57]. We have used this protocol to determine OTOC in discrete drive protocol in [65]. Plugging this choice of Lorentzian time periods in A.8 we get

$$\sqrt{\alpha^2 - 4\beta\gamma} = \frac{4\theta}{(T_0 + T_1)}, \tag{A.11}$$

$$\alpha = 0, \; \beta = -\gamma = \frac{2\theta}{T_0 + T_1}.$$

Note that, the Floquet Hamiltonian for this choice reduces to[20]

$$e^{-i(\tau_0 + \tau_1)H_{\text{eff}}} = e^{2\theta\left(L_1 - L_{-1} + \bar{L}_1 - \bar{L}_{-1}\right)}. \tag{A.12}$$

Interestingly, this Hamiltonian also annihilates the boundary state $|B\rangle$[21] apart from the vacuum. Thus this Hamiltonian can not distinguish between vacuum and boundary state.

## B   Derivation of the solution to the tangent equations

In this appendix, we show the derivation of the solutions to the tangent equations (7)-(9) for all three phases in more detail. We begin by rewriting the bulk extension of the boundary effective Hamiltonian, which is given by

$$H_b = (-\alpha z + 2\beta zX)\partial_z + \left(-\alpha X - \beta z^2 + \beta(X^2 - \tau^2) + \gamma\right)\partial_X + (-\alpha\tau + 2\beta X\tau)\partial_\tau. \tag{B.1}$$

---

[20]One may wonder how we get $\alpha^2 - 4\beta\gamma > 0$ in heating phase. This is due to the fact, when we write $U_{\text{eff}} = e^{-(\tau_0 + \tau_1)H_{\text{eff}}}$, upon analytic continuation to Lorentzian time, we can write the evolution operator with $H_{\text{eff}} = i\alpha L_0 + i\beta L_1 + i\gamma L_{-1}$. This shift of $\alpha, \beta, \gamma \to i\alpha, i\beta, i\gamma$ changes the sign of the Casimir and hence the sign of $\alpha^2 - 4\beta\gamma$.

[21]By definition, $(L_n - \bar{L}_{-n})|B\rangle = 0$.

Subsequently, we rewrite the corresponding tangent equation as the follows:

$$\frac{dz(s)}{ds} = 2\beta z\left(X - \frac{\alpha}{2\beta}\right) = 2\beta z x, \tag{B.2}$$

$$\frac{d\tau(s)}{ds} = 2\beta \tau\left(X - \frac{\alpha}{2\beta}\right) = 2\beta \tau x, \tag{B.3}$$

$$\frac{dx(s)}{ds} = \beta\left[\left(X - \frac{\alpha}{2\beta}\right)^2 - \tau^2 - z^2 - \left(\frac{\alpha^2}{4\beta^2} - \frac{\gamma}{\beta}\right)\right] = \beta\left(x^2 - \tau^2 - z^2 - d\right), \tag{B.4}$$

where, $x = (X - \frac{\alpha}{2\beta})$ and $d = (\frac{\alpha^2}{4\beta^2} - \frac{\gamma}{\beta})$. Then using (B.2) and (B.3) one can first solve, $\frac{dz}{d\tau} = \frac{z}{\tau}$ to get $z = c_1 \tau$. Now, the other two solutions can be obtained straight forwardly by substituting $z = c_1 \tau$ back in the above equations and then solving the following (for each phase):

$$\frac{du}{ds} = \beta(u^2 - d), \tag{B.5}$$

$$\frac{dv}{ds} = \beta(v^2 - d), \tag{B.6}$$

written in terms of redefined variables $u = x + i\delta\tau$ and $v = x - i\delta\tau$ with $\delta^2 = 1 + c_1^2$.

### For Non-heating Phase ($d > 0$):

First, one can easily check that solving (B.5), one gets: $\frac{u - \sqrt{d}}{u + \sqrt{d}} = c_2 e^{2\sqrt{d}\beta s}$, where $c_2$ is a complex number. Next, after redefining the following $c_2 = Re^{i\theta}$, $s + \frac{\log[R]}{2\sqrt{d}\beta} \to s$ and $\frac{\theta}{2\sqrt{d}\beta} \to \theta$, the solution can be written as $u = -\sqrt{d} \coth(\beta\sqrt{d}(s + i\theta))$ and similarly $v = -\sqrt{d} \coth(\beta\sqrt{d}(s - i\theta))$. From $u = x + i\delta\tau$ and $v = x - i\delta\tau$ one can then obtain the final form of the solution in (10).

### For heating phase ($d < 0$):

Approaching similarly as that of the non-heating case, one solves (B.5) with $d < 0$, to obtain $\tan^{-1}(\frac{u}{\sqrt{d}}) = c_2 + \sqrt{d}\beta s$, where $c_2$ can again be redefined as $c_2 = x + iy$, at first and then just like the previous case, one can redefine $s + \frac{x}{\sqrt{d}\beta}$ and $\frac{y}{\sqrt{d}\beta}$ as $s$ and $\theta$ respectively.

The solutions to (B.5) and (B.6) for this case then read, $u = \sqrt{d} \tan(\beta\sqrt{d}(s + i\theta))$ $v = \sqrt{d} \tan(\beta\sqrt{d}(s - i\theta))$. $x$ and $\tau$ can then be found out using $u = x + i\delta\tau$ and $v = x - i\delta\tau$ and $z$ using $z = c_1\tau$. Hence, one obtains the solution in (14) and the corresponding metric.

### At the phase boundary ($d = 0$):

The solution to (B.5) and (B.6) reads $u = v = -\frac{1}{\beta s + c_2}$. Again, considering $c_2 = x + iy$ and redefining $s + x$ and $\frac{y}{\beta}$ as $s$ and $\theta$, one can find the solutions in (17).

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
