# Peer review of "Brane Detectors of a Dynamical Phase Transition in a Driven CFT"

_SciPost Physics, doi:SciPost Phys. 15, 202 (2023)_

## Round 1 · Referee Report · Anonymous (Referee 1) · 2023-3-12

Report

This paper studies non-heating to heating transitions in 2d CFTs using holographic descriptions involving gravity+brane systems. A general effective Hamiltonian mapped to its AdS3 representation (sec.2) leads to bulk curves generated correspondingly, parametrized by some stroboscopic time s, and corresponding coordinate solutions (in the non-heating, heating and transition phases). Appropriate codim-1 probe brane embeddings via these coordinate solutions (sec.3) then shows a difference in the brane on-shell action regarded as free energy. Further studies are carried out via end-of-world branes as diagnostics: these reveal various detailed differences (stemming from EOW-brane extrinsic curvature effects here). The authors then study boundary 2-point correlation functions (sec.4) via bulk geodesic approximations and 2d CFT on the appropriate boundary slice, which match in appropriate regimes. Subsequently they study OTOCs.

I find the paper interesting, comprehensive in what it studies, and worth publication.

I have a few questions, some of which appear listed in the Discussion section:

(1) The class of Hamiltonians considered is not the most general: it would be interesting to understand these generalizations. It would seem some of these will have nontrivial time-dependence, which may be interesting for various purposes.

(2) The AdS2 slicing plays crucial roles: this appears special. Naively, general foliations might suggest the 2d slices being conformally AdS2 (which would be rather different). These choices of AdS2 slices might dovetail with particular "static gauge" choices for the probe branes (in eqs.3.11-3.12 via $\ \sigma_0=s,\ \sigma_1=\theta$), but it's not clear to me if this is true in general, going from the considered boundary metrics to the bulk. I'm wondering if this links back to (1) above.

(3) Besides correlation functions, an obivous probe of such phenomena is entanglement entropy. In the holographic context, the corresponding RT/HRT surfaces will amount to geodesics, but with various differences in the thinking. It may be interesting to explore this as well as the generalized entropy (via appropriate quantum extremal surfaces). There may also be useful things to gain from studying double holography.

Additionally, for completeness, perhaps it will be useful to add the following in whatever way the authors deem fit:

(a) a little more elaboration on the AdS3 representations of the CFT Hamiltonian, the three phases and the sign of d.

(b) some detail on the coordinate solutions to eqs.2.7-2.9, possibly in an Appendix (perhaps also including a short review of this, e.g. from ref.[60]?).

---

## Round 1 · Referee Report · Anonymous (Referee 2) · 2023-5-23

Report

Authors consider quantum dynamics in 2d CFTs generated by action of a unitary build from the global SL(2,R) sub-algebra (two copies) on the vacuum state. Physically, this can be interpreted as an inhomogeneous quantum quench or a periodic Floquet drive. This has been studied in various examples and, depending on the parameters of the evolving Hamiltonian, a dynamical transition between a heating and non-heating phases was observed.

The main contribution of this work is a holographic proposal or interpretation of such proces in holographic 2d CFTs (since these states and operators are completely universal, there is no distinction between holographic and non-holographic 2d CFTs). Authors identify the SL(2,R) generators in the effective Floquet Hamiltonian with the SL(2,R) generators in AdS3 and find a local frame in which this operator generates time translations “s”. This is interesting (new or standard?) and may give some hints on local frames in the bulk (e.g. see 2211.16512 [hep-th]).

The crucial part of the holographic picture is played by the “EOW” branes that are chosen as constant mean curvature slices of AdS3. Authors compute gravity action (in the probe limit as well as with back-reaction) and argue that it can distinguish between the different phases.

I find the paper interesting and worth publishing after addressing a few minor comments:

  1. Acting with the unitary on some more non-trivial highest weight state would be much more interesting and would involve more input about holographic CFT (large c, sparseness). Was it too difficult to analyse?
  2. The action with the probe brane as well as the CMC slices played a key role in arXiv:2104.00010v2 [hep-th]. In fact the general construction is quite similar and, modulo some boundary-terms one could interpret the bulk computation that distinguishes the phases as holographic path-integral complexity… Maybe worth exploring or commenting on the connection.
  3. On a related note, after back-reaction one may think about the bulk setup as an example of the AdS/BCFT framework. Is there any sign of this from the CFT perspective (given that its just a quench in ordinary CFT without any boundaries)?
  4. Above (2.12) it should be “c_1 = tan φ” and not $\phi_1$. Btw, where does this come from? How do they know that this constant should become one of the “bulk coordinates”.
  5. Maybe some Hamiltoni-Jacobi perspective could be useful for the previous question?
  6. What does the dot “.” in (A.1) mean?

---

## Round 2 · Author Response

We thank both the Referees for their insightful remarks. We have added several comments as well as a couple of new paragraphs in the Discussion section to further emphasize these points. We are also currently actively working on related aspects, which we hope to report soon.

To further facilitate visibility, we are enlisting here our responses to each point.

——————————————————————————————————— Referee 2:

Q1- Acting with the unitary on some more non-trivial highest weight state would be much more interesting and would involve more input about holographic CFT (large c, sparseness). Was it too difficult to analyse?

Response —- Indeed, the analysis would be more interesting with an excited state. However, there are two potential challenges in the Holographic picture: Representing the excited state in the bulk and solving the equations of the curve. The latter issue seems more technically involved.

Even for the vacuum, taking a more general form of the Hamiltonian, we find that the equations of the curve are considerably more difficult to solve. This is true also for the case when the state is obtained by a unitary generated by the sl(q, R) generators, where the bulk dual geometries are Banados geometries. We are currently working on this problem and hopefully we will find an interesting aspect to report soon. This deserves a separate study altogether.

** Modification to draft v1: We have commented about this possibility in the second paragraph of the Discussion section.

Q2-The action with the probe brane as well as the CMC slices played a key role in arXiv:2104.00010v2 [hep-th]. In fact the general construction is quite similar and, modulo some boundary-terms one could interpret the bulk computation that distinguishes the phases as holographic path-integral complexity… Maybe worth exploring or commenting on the connection.

Response -— This is a very interesting comment. It is indeed feasible that a precise connection exists between these two apparently disparate scenarios. Furthermore, the addition of back-reacting branes deserves to be explored on their own for several reasons, including one that is mentioned in the response to Q3 below.

** Modification to draft v1: We feel this requires a full-fledged analyses on its own and therefore we have added paragraph 5 discussing this possibility in our Discussion section.

Q3. On a related note, after back-reaction, one may think about the bulk setup as an example of the AdS/BCFT framework. Is there any sign of this from the CFT perspective (given that its just a quench in ordinary CFT without any boundaries)?

Response — The back-reacted description should indeed correspond to a BCFT framework. This is expected to be more general from the CFT framework considered in the first part of the paper. However, it emerges as a natural generalization, especially in the holographic description, and in the presence of the branes in a global AdS patch, the brane on-shell action is expected to yield the so-called boundary entropy. This boundary entropy is subsequently expected to capture the phase transition. Work along this direction is underway. This aspect is intimately related to the point above and will be explored in the near future.

Q4-Above (2.12) it should be “c_1 = tan φ” and not ϕ1. Btw, where does this come from? How do they know that this constant should become one of the “bulk coordinates”. Q5- Maybe some Hamilton-Jacobi perspective could be useful for the previous question?

Response -- We again thank the Referee for pointing out the typo in coordinate "φ". Regarding the constant being the bulk coordinate, the idea is that the constant of integration along a curve should be interpreted as a coordinate which is "orthogonal" to the curve, i.e. it does not change along the curve. For example, the equation of a circle around the origin in two dimensions would be parametrized as x = Rcos(theta) and y = Rsin(theta). Here R would be a constant along the circle but should be thought of as another coordinate, along with theta, from the perspective of the two-dimensional space. Of course, this is not a unique choice of coordinates, but it was a convenient one since the metric in these coordinates takes a simple form.

We have not thought about the problem from the Hamilton- Jacobi perspective. It may be useful and interesting and maybe worth exploring further, however, for us it did not seem essential.

Q6- What does the dot “.” in (A.1) mean?

Response — The "." in A1 is a typo. We will be changing it in the revised version.

—————————————————————————————————————————————

Referee 1:

Q1-The class of Hamiltonians considered is not the most general: it would be interesting to understand these generalizations. It would seem some of these will have nontrivial time-dependence, which may be interesting for various purposes.

Response —- Indeed this is a crucial point that deserves more attention. There are several classes of potential generalisations.

For example, we are currently exploring the bulk dual of a Hamiltonian that is constructed out of SL_q(2,R) generators. This appears technically involved. We hope to devise a new method in the future to overcome this problem. Undoubtedly, there are more and different types of generalisations that are also worth pursuing systematically. We hope to report on some of them in near future.

** Modification to draft v1: We have updated paragraph 2 in the Discussion section, emphasising this point.

Q2- The AdS2 slicing plays crucial roles: this appears special. Naively, general foliations might suggest the 2d slices being conformally AdS2 (which would be rather different). These choices of AdS2 slices might dovetail with particular "static gauge" choices for the probe branes (in eqs.3.11-3.12 via σ0=s, σ1=θ), but it's not clear to me if this is true in general, going from the considered boundary metrics to the bulk. I'm wondering if this links back to (1) above.

Response — Indeed, the AdS_2 slices are special and it is not completely clear to us how generic slices will perceive the phase transition. Our primary motivation was to capture the phase transition which, as rightly pointed out, is done by the static gauge branes. More and independent work is needed to address this issue completely.

** Modification to draft v1: We have updated paragraph 6 in the Discussion section, emphasising this point explicitly.

Q3: Besides correlation functions, an obvious probe of such phenomena is entanglement entropy. In the holographic context, the corresponding RT/HRT surfaces will amount to geodesics, but with various differences in the thinking. It may be interesting to explore this as well as the generalized entropy (via appropriate quantum extremal surfaces). There may also be useful things to gain from studying double holography.

In the set-up we consider in this paper, sl(2,R) drive of the vacuum, equal time correlation functions do not evolve in time. So EE is not an interesting observable in our case. However for more general drives, it will indeed be very useful and interesting to compute EE and other probes to better decipher the connection between the driven CFT systems and the Doubly-Holographic systems. Towards this, some related literature already exists that may be of direct relevance: e.g. 2109.00079. In fact, one motivation of our current work is to lay out a basic and elementary platform on which we intend to address these questions in near future. This can further broaden the scope of such Doubly-Holographic models.

** Modification to draft v1: We have added a new paragraph at the end of the Discussion section further elucidating this point.

Additionally, for completeness, perhaps it will be useful to add the following in whatever way the authors deem fit: (a) a little more elaboration on the AdS3 representations of the CFT Hamiltonian, the three phases and the sign of d.

  1. some detail on the coordinate solutions to eqs.2.7-2.9, possibly in an Appendix (perhaps also including a short review of this, e.g. from ref.[60]?).

** Modification to draft v1: We have updated the draft with a new appendix (Appendix B)where we have elaborated on the details of derivation of the coordinate solutions.

System Message: WARNING/2 (<string>, line 111)

Title underline too short.

** Modification to draft v1: We have updated the draft with a new appendix (Appendix B)where we have elaborated on the details of derivation of the coordinate solutions.
* * *
We hope that the above adequately addresses both Referees' points.

---

## Round 2 · List of Changes

Here is a list of changes that we have made in our re-submission:

1. A number of typos have been fixed, including one that was pointed out by Referee 2.

2. We have added more relevant references, especially, eg ref [69,70,71].

3. We have added a new appendix B, where more explicit details related to solving the equations of motion for the integral curve are provided.

4. To emphasize several interesting and important points mentioned by both the Referees, we have considerably modified the Discussion section. In particular, we have implemented the following changes:

i) Updated the following paragraphs: paragraph 2, 6

ii) Added the following new paragraphs: paragraph 5, 8 and 9.

---

## Editorial Decision

published